# Genetically encoded photocatalytic protein labeling enables spatially-resolved profiling of intracellular proteome

Fu Zheng[1], Chenxin Yu[2,3], Xinyue Zhou[2] & Peng Zou [1,2,4,5] ✉

Mapping the subcellular organization of proteins is crucial for understanding their biological functions. Herein, we report a reactive oxygen species induced protein labeling and identification (RinID) method for profiling subcellular proteome in the context of living cells. Our method capitalizes on a genetically encoded photocatalyst, miniSOG, to locally generate singlet oxygen that reacts with proximal proteins. Labeled proteins are conjugated in situ with an exogenously supplied nucleophilic probe, which serves as a functional handle for subsequent affinity enrichment and mass spectrometry-based protein identification. From a panel of nucleophilic compounds, we identify biotin-conjugated aniline and propargyl amine as highly reactive probes. As a demonstration of the spatial specificity and depth of coverage in mammalian cells, we apply RinID in the mitochondrial matrix, capturing 477 mitochondrial proteins with 94% specificity. We further demonstrate the broad applicability of RinID in various subcellular compartments, including the nucleus and the endoplasmic reticulum (ER). The temporal control of RinID enables pulse-chase labeling of ER proteome in HeLa cells, which reveals substantially higher clearance rate for secreted proteins than ER resident proteins.

Within highly compartmentalized eukaryotic cells, the subcellular localization of proteins is crucially linked to their biological functions. This is most readily observed in secreted proteins, which constantly traffic through the endoplasmic reticulum (ER)-Golgi apparatus-plasma membrane axis[1]. In signal transduction pathways or stress response pathways, activated protein factors often translocate from cytoplasm[2] or mitochondria[3] into the nucleus to initiate the transcription of effector genes. Thus, our understanding of protein function would require knowledge of not only the abundances and activities of individual proteins, but also their spatial arrangements, ideally at the proteome level and in the native cellular context[4].

To profile the subcellular organization of proteome, a number of spatial-specific chemical labeling techniques have been developed over the past decade, which often capitalize on genetically targetable enzymes (e.g., APEX[5,6], BioID[7], TurboID[8]) that catalyze the formation of short-lived and highly reactive intermediates in live cells (e.g., biotin-conjugated phenoxyl radicals[5,6] or biotinyl 5′-adenylate[7,8]). These intermediates react with proteins in close proximity to their source of generation, thus achieving high spatial specificity of labeling. Proximity labeling enzymes have been applied to investigate the proteome of many subcellular compartments, including the mitochondria[9], endoplasmic reticulum[10], primary cilia[11], etc.

However, the requirement of using $H_2O_2$ in APEX labeling may cause toxicity to living samples[12]. Cellular expression of constitutively active TurboID would lead to high background biotinylation of endogenous proteins, which may interfere with cellular physiology and

[1]College of Chemistry and Molecular Engineering, Synthetic and Functional Biomolecules Center, Beijing National Laboratory for Molecular Sciences, Key Laboratory of Bioorganic Chemistry and Molecular Engineering of Ministry of Education, Peking University, Beijing 100871, China. [2]Academy for Advanced Interdisciplinary Studies, Peking-Tsinghua Center for Life Sciences, Peking University, Beijing 100871, China. [3]College of Chemistry and Chemical Engineering, Lanzhou University, Lanzhou 730000, China. [4]PKU-IDG/McGovern Institute for Brain Research, Peking University, Beijing 100871, China. [5]Chinese Institute for Brain Research (CIBR), Beijing 102206, China. ✉e-mail: zoupeng@pku.edu.cn

cause cytotoxicity. These problems have motivated us to search for a new proximity labeling method, where the activation avoids toxic $H_2O_2$ and could be controlled by light trigger, to minimize the impact on cell physiology.

Recently, several small-molecule-based photocatalytic protein proximity labeling methods have been reported. For example, μMap[13] and μMap-Red[14] use transition metal-centered photocatalyst to convert diazirine or phenyl-azide into highly reactive carbene or nitrene intermediates, which are covalently conjugated to nearby proteins. These methods have been used to map the interactome of cell-surface proteins. Another method, termed CAT-prox[15], uses iridium-centered photocatalyst to liberate reactive quinone methide from its azidobenzyl-protected precursor, which subsequently reacts with nucleophilic residues of neighboring proteins. Alternatively, a photocatalyst could be used to generate singlet oxygen in situ to oxidize electron-rich residues on nearby proteins, which are then captured with a nucleophilic probe. For example, a combination of dibromofluorescein-Hoechst photocatalyst and biotin-PDA probe have been applied for labeling nuclear proteome[16]. LUX-MS applies antibody- or drug-conjugated small-molecule singlet oxygen generator and biotin-hydrazide probe to decode ligand-receptor interactions and the proteomes on cell surface[17].

However, a common drawback of small-molecule-based proximity labeling methods is the difficulty of achieving highly specific subcellular localization of the photocatalysts. Antibody-conjugates could target specific bait proteins, but have been limited to cell surface labeling due to lack of membrane permeability. While the intracellular targeting of photocatalysts has been demonstrated for the mitochondria and the nucleus, targeting other subcellular compartments has remained challenging, which is common problem associated with many small-molecule-based techniques. In contrast, genetically encoded methods could more readily achieve subcellular targeting of protein-based photocatalyst through fusion with protein markers or signal peptides.

MiniSOG is a photocatalytic protein that could be genetically targeted to various subcellular compartments[18]. Upon blue light illumination, miniSOG generates singlet oxygen, which is capable of oxidizing a wide range of biomolecules including nucleic acids and proteins. Notably, miniSOG has been used for mapping the subcellular transcriptome (CAP-seq)[19] and for probing protein-protein interactions[20], which demonstrates its high spatial specificity in labeling local biomolecules. Yet the subcellular proteome-wide identification by miniSOG has not been reported.

Herein, we report a miniSOG-based light-activatable proximity labeling method for profiling subcellular proteomes with minute-level turn-on kinetics, excellent labeling efficiencies, and high spatial specificity in various organelles. Our method, called reactive oxygen species (ROS)-induced protein labeling and identification (RinID), capitalizes on miniSOG-mediated photo-oxidation of proximal proteins. Photo-oxidized protein intermediates are intercepted with a nucleophilic probe and subsequently enriched for mass spectrometry analysis. We have screened a panel of nucleophilic compounds and identified biotin-conjugated aniline and propargylamine as highly reactive probes. Application of RinID to the mitochondrial matrix identifies 477 proteins with 94% mitochondrial specificity, which compares favorably to previously reported methods. RinID can also be applied in other subcellular compartments (e.g., nucleus and ER), thus demonstrating its broad applicability and capability of complementing other proximity labeling methods. We further apply RinID to pulse-chase labeling of proteins in the ER lumen of Hela cells, which reveals a broad distribution of protein clearance rate in the secretory pathway, with secreted proteins turning over substantially faster than ER resident proteins.

## Results

### MiniSOG photo-oxidizes proteins at histidine residues

Proteins are prone to be oxidized by reactive oxygen species (ROS)[21]. For example, the imidazole ring of histidine could be oxidized into 2-oxo-imidazole in the presence of singlet oxygen[22]. In cells, extensive oxidation of a protein could hamper its enzymatic activity or interaction with other biomolecules, thus turning off its function. This feature has been leveraged in chromophore-assisted light inactivation (CALI[23]) strategy to achieve selective photo-ablation of specific proteins in live cells. miniSOG, an engineered flavin-binding protein, has been employed in CALI experiments as a protein fusion tag[24], which generates singlet oxygen upon blue light illumination. In this study, we aim to repurpose miniSOG for ROS-induced proximity-dependent proteome labeling and identification (RinID). We propose to capture the protein photo-oxidation intermediates in situ with amine-based nucleophilic probes functionalized with an affinity purification handle. We reason that, due to the short lifetime (<0.6 μs) and limited diffusion radius (~70 nm) of singlet oxygen[25], such labeling reaction occurs only proximal to miniSOG, and the labeled proteins could be subsequently enriched and identified through mass spectrometry-based proteomic analysis (Fig. 1A).

We started by testing miniSOG-mediated photo-oxidation in vitro with a model protein, bovine serum albumin (BSA, PDB: 4F5S), with biotin-conjugated alkyl amine (biotin-PEG-$NH_2$, probe 1) as the nucleophilic probe (Fig. 1B, Supplementary Fig. 1A–B). In the presence of 100 μM purified miniSOG and 20 mM biotin-PEG-$NH_2$, BSA in phosphate buffer saline solution (pH 7.3) was illuminated with 460–470 nm blue LED at the mild intensity of 19 mW·$cm^{-2}$ for 30 min at room temperature. Western blot analysis showed successful biotinylation of BSA (Fig. 1B). In negative controls omitting either miniSOG or light illumination, the biotinylation signal was substantially reduced. We attributed the low biotinylation background in the absence of miniSOG to the residual serum-derived photosensitizer impurities in the BSA sample. We repeated the labeling with another nucleophilic probe, propargyl amine (PA, probe 6, Supplementary Fig. 1A) and obtained similar results (Supplementary Fig. 1C). We repeated the above in vitro labeling by probe 1 with a purified protein, sortase A, and obtained similar results. No background labeling in the negative controls omitting miniSOG or light illumination was observed (Supplementary Fig. 1D). We also noticed a change in Coomassie brilliant blue (CBB) staining pattern when protein samples containing miniSOG were illuminated with blue light, which suggested protein crosslinking mediated with singlet oxygen[26–28]. Together, the above characterizations demonstrate that miniSOG is capable of labeling proteins with amine-conjugated probes in a blue light-dependent manner.

To understand the labeling mechanism, we searched for the amino acid residues of photo-oxidation and probe 6 conjugation by mass spectrometry. Photo-oxidized BSA sample was proteolytically digested into peptide fragments and analyzed by liquid chromatography-tandem mass spectrometry (LC-MS/MS). Among the five amino acid residues (histidine, tyrosine, tryptophan, cysteine, and methionine) that are commonly oxidized by singlet oxygen[21], the photo-oxidation products of histidine were most readily detected on the mass spectrometry, with observed mass shifts of 31.990 Da and 69.022 Da matching the transformation of imidazole ring into 5-hydroxy-1,5-dihydro-2-oxoimidazole and 5-propargylamino-1,5-dihydro-2-oxoimidazole (Fig. 1C, Supplementary Fig. 1E). This observation is consistent with a recent report that amine probe 1-methyl-4-arylurazole could react with photocatalytically oxidized histidine residue[29]. In both cases, singlet oxygen reacts with the imidazole ring to form an endoperoxide intermediate, which undergoes nucleophilic addition at the C4 position by either water or an amine probe, generating C–O and C–N bond, respectively. Consistent with this mechanism, the m/z of both products were identified at solvent-exposed His18 and His378 sites of BSA following blue LED irradiation in the presence of probe 6 (Fig. 1C). In addition, although we have also identified the oxidized products of tryptophan, tyrosine, and methionine, we have failed to detect their nucleophilic addition products with probe 6 (Supplementary Data 1, Supplementary Fig. 1F–H).

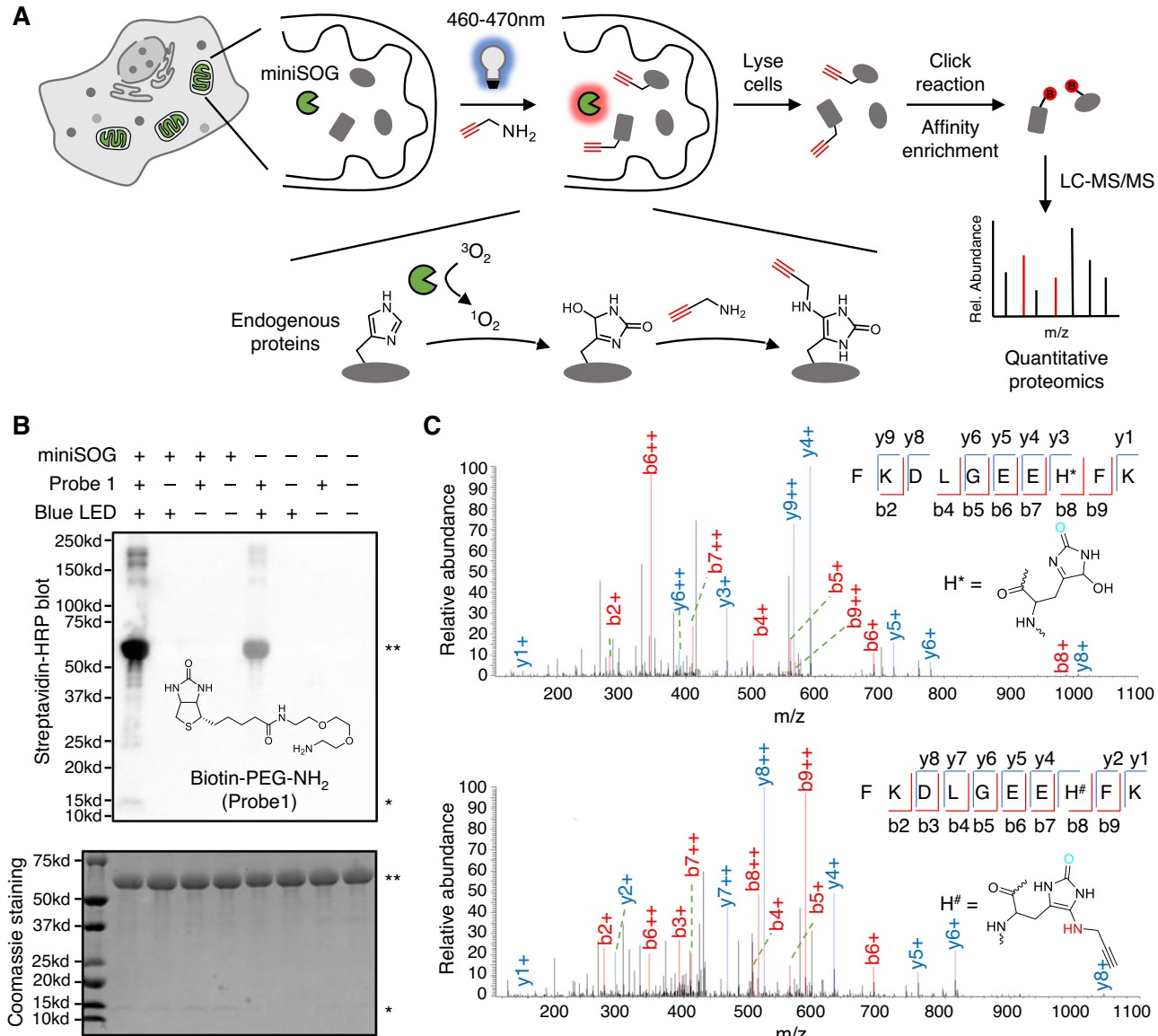

**Fig. 1 | Experimental scheme and in vitro characterization of ROS-induced proximity-dependent protein labeling and identification (RinID). A** Scheme of RinID workflow. **B** Western blot and SDS-PAGE (4–20% gradient) analysis of miniSOG-mediated photo-oxidation of the model protein, bovine serum albumin (BSA) with 20 mM Biotin-PEG-NH2 probe. Samples were illuminated with blue LED at 19 mW·cm⁻² for 30 min. * miniSOG, ** BSA. **C** MS/MS spectra of a representative peptide with histidine residue oxidized to 5-hydroxy-1,5-dihydro-2-oxoimidazole (top) and 5-propargylamino-1,5-dihydro-2-oxoimidazole (bottom).

We conclude from the above data that miniSOG could mediate the photocatalytic protein conjugation with amine probes. Due to the competition from water and other nucleophilic species in living cells, probe labeling needs to be optimized for efficient protein capture.

### Application of RinID to multiple subcellular compartments

We next sought to achieve miniSOG-mediated protein labeling in live cells. We constructed human embryonic kidney 293T (HEK293T) cell lines targeting miniSOG to various subcellular compartments, including the cytoplasm, membrane-delimited organelles (e.g., mitochondrial matrix, ER), and membraneless condensates (e.g., stress granule) (Fig. 2A). Meanwhile, we prepared a panel of biotin-conjugated amine probes (**1, 2, 4, 5**), including primary alkyl amine, aniline, and hydrazide, which differ in nucleophilicity, steric hindrance, and basicity (Fig. 2B). For comparison, we also included biotin-conjugated phenol (probe **3**), the commonly used substrate for APEX[5,6]. Since propargylamine has been used in CAP-seq to capture photo-oxidized guanosine, we added this probe (**6**) in our list of candidates as well. We tested the

labeling efficiency of these probes both in the HEK293T cell lysate and live cells stably expressing cytoplasmic targeted miniSOG. For probe **6**, copper(I)-catalyzed alkyne-azide cycloaddition (CuAAC) click reaction with biotin-azide was performed to install the biotin moiety prior to Western blot analysis. Whereas a similar labeling efficiency was observed for probe **6** and probes **1-4** in the cell lysate (Supplementary Fig. 2A), only probe **6** (propargylamine) and probe **2** (biotin-aniline) yielded strong labeling signal in living cells (Supplementary Fig. 2B). We speculated that probes **1, 3**, and **4** may have limited permeability through the cell membrane, while probe **5** lacks the ability to capture photo-oxidized protein intermediates.

We further optimized probe concentration and blue light illumination time for RinID labeling. The excellent water solubility of probe **6** enables it to be supplied at higher concentration (e.g., 10–20 mM), which favorably competes with water and endogenous nucleophiles at intercepting the miniSOG-mediated protein photo-oxidation intermediate. Using mitochondrial matrix-targeted miniSOG as a model and Western blot signal intensity as the readout, we determined

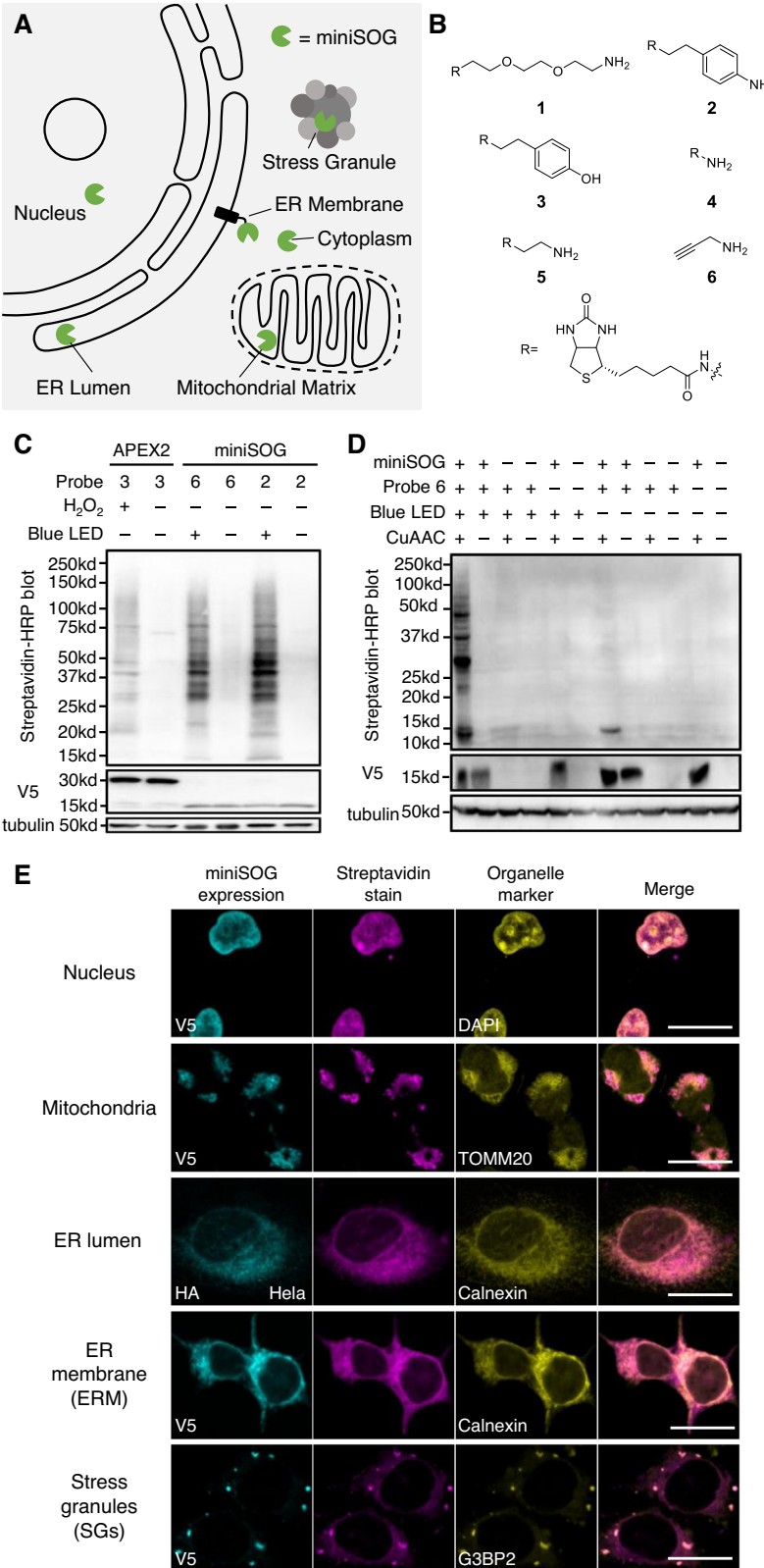

**Fig. 2 | Application of RinID at different subcellular localizations in living cells.**
**A** Subcellular targeting of miniSOG at different organelles. **B** Chemical structures of
probes used in this study. **C, D** Western blot analysis (12% SDS-PAGE) comparing of
mitochondrial matrix-targeted RinID labeling and APEX2 (miniSOG or APEX was
fused to V5 tag) (**C**), and comparing RinID labeling against control experiments
omitting (blue light illumination, probe **6**, miniSOG, and/or CuAAC) (miniSOG was
fused to V5 tag) (**D**). **E** Confocal fluorescence images of cultured mammalian cells
labeled with RinID in different organelles. Scale bars: 20 μm. HEK293T cells were
used unless otherwise noted. miniSOG was fused to V5 or HA tag.

20 mM probe **6** (Supplementary Fig. 3A) for 15 min (Supplementary Fig. 3B) as the optimal condition. The overall biotinylation signal with 20 mM probe **6** was comparable to 5 mM probe **2** in this cell line (Fig. 2C). Notably, low levels of biotinylation background could be observed in negative controls omitting miniSOG or light illumination (Fig. 2D), which we attributed to the presence of native photosensitizers (e.g., FMN) and naturally oxidized proteins in cells. To remove this background signal, ratiometric quantitative mass spectrometry experiments with proper controls should be performed.

We benchmarked the labeling efficiency of our method with APEX2, which has been widely used for proximity labeling. In the mitochondrial matrix, miniSOG-mediated protein labeling with probes **2** and **6** are both substantially higher than APEX2-mediated labeling with biotin-phenol, even when miniSOG was expressed at a lower level than APEX2 (Fig. 2C). However, it should be noted that miniSOG-mediate labeling typically requires light illumination for 15 min, whereas APEX2 requires only 1 min or less. Thus, when studying highly dynamic biological processes, such as G-protein coupled receptor signaling, APEX2 is still recommended for its fast reaction kinetics[30,31]. Taken together, the above analysis established **2** and **6** as suitable probes for RinID. We considered probe **6** as a more cost-effective probe due to its commercial availability, which we used for all subsequent experiments.

To evaluate the spatial specificity of miniSOG-mediated protein labeling, we performed immunofluorescence imaging of cell samples labeled with probe **6** (Fig. 2E). Following photo-oxidation, cells were fixed and permeabilized with cold methanol. Biotinylation signal was detected by staining cells with streptavidin-conjugated fluorophores, while the localizations of miniSOG and the morphology of relevant organelles were visualized via antibody staining (or DAPI staining in the case of nucleus). Confocal fluorescence microscopy reveals good co-localization between biotinylated proteins and organelle markers, thus demonstrating the high spatial specificity of our method. In wild-type HEK293T cells lacking miniSOG and in negative controls omitting probe **6** or light illumination, labeling was almost undetectable. However, we did notice the presence of a low biotinylation background that permeated throughout the cytoplasm and nucleus (Supplementary Fig. 4). This background was reminiscent of our previous observation in Western blot analysis, which was likely caused by endogenous photosensitizer and oxidized proteins. Collectively, RinID could label subcellular proteomes with high spatial specificity within 15 min at various membrane-bound and membraneless organelles in different cell lines.

### Profiling mitochondrial matrix proteome with RinID

We then evaluated the specificity and coverage of RinID with quantitative mass spectrometry (MS)-based proteomic profiling. HEK293T cells expressing mitochondrial matrix-targeted miniSOG were incubated with 20 mM probe **6** and illuminated with blue LED at 30 mW·cm$^{-2}$ for 15 min (Fig. 3A). Following light illumination, cells were collected and lysed, and the lysate was reacted with biotin-conjugated azide via click reaction. Thereafter, biotinylated proteins were captured by streptavidin-coated agarose beads. Successful enrichment was confirmed by SDS-PAGE and silver staining (Supplementary Fig. 5A–B).

As mentioned above, background labeling should be carefully removed via quantitative MS experiments. For this purpose, we designed two negative controls and applied stable isotope dimethyl labeling strategy to quantitatively measure the ratios of protein abundance between samples. While one control omitted blue LED irradiation to account for probe **6** labeling on native oxidized proteins, the other control used wild-type HEK293T cells lacking miniSOG to eliminate background protein labeling induced by endogenous photosensitizers (Fig. 3A, B). For each set of experiments (+/− blue LED or +/− miniSOG), two biological replicates were performed. Both labeled samples and control samples went through the same enrichment

workflow and subsequently digested by trypsin. The resulting peptides were treated with isotope-encoded formaldehyde (heavy D$^{13}$CDO for labeled samples versus light HCHO for control samples) and NaBH$_3$CN to methylate their −NH$_2$ groups, leading to mass shifts of 34.0631 Da versus 28.0313 Da, respectively. The heavy and light samples were mixed and analyzed by LC-MS/MS for peptide identification and abundance determination.

A total of 1634 and 1882 proteins were identified and quantified in both replicates for "+/− blue LED" and "+/− miniSOG" datasets, respectively. For each dataset, proteins were ranked by their averaged H/L ratios, and the cut-off ratios were determined with receiver operator curve (ROC) analysis (Supplementary Fig. 5C, Supplementary Data 2). For ROC analysis, 1132 proteins in human MitoCarta 3.0[32], a well-established human mitochondrial proteome database, were defined as the 'true positive' list (Supplementary Data 2). The 'false positive' list consisted of 2403 proteins that were previously annotated as 'false positives' in the work of mitochondrial matrix TurboID[8] (Supplementary Data 2). The cut-off log$_2$H/L ratios were set at 1.21 and 1.24, yielding 611 and 556 enriched proteins for "+/− blue LED" and "+/− miniSOG" datasets, respectively. The overlap of these two lists contained 477 proteins, which was defined as our mitochondrial proteome (Supplementary Data 2).

Notably, this protein inventory has exceptionally high mitochondrial specificity, with 94% (450 out of 477) of proteins listed in the MitoCarta 3.0, which is higher than previously reported proximity labeling methods, including APEX (92%)[5], TurboID (59%)[8], and small molecule photosensitizer-based CAT-Prox (70%)[15] (Fig. 3C). In terms of sub-mitochondrial specificity, 63 and 30% of our RinID dataset are mitochondrial matrix and inner membrane proteins, respectively (Fig. 3D). To further examine the coverage and the spatial specificity of RinID, we use the electron-transport chain complexes as a model, whose membrane topology has been well resolved through structural biology studies. Figure 3E and Supplementary Data 2 show that the majority of protein components identified by RinID are exposed to the mitochondrial matrix, where biotinylation occurs. The coverage of mitochondrial proteins by RinID is similar to that of APEX (495 proteins) and TurboID (311 proteins), with an overlap of 128 proteins, almost all of which (126 proteins) are annotated in the MitoCarta 3.0 database. In addition, 151 proteins are uniquely identified by RinID (Fig. 3F), including 128 mitochondrial proteins (85%). This difference in coverage may arise from the preferences of three methods toward different amino acid residues: whereas APEX2 and TurboID favor tyrosine and lysine, respectively, RinID targets histidine. In addition, variations in the labeling protocols (photocatalytic vs. enzymatic) and the quantitative proteomics analysis methods could also contribute to the difference in coverage. Taken together, the above comparisons indicate that RinID offers exceptional spatial specificity and good coverage, and could complement existing methods.

### Profiling ER membrane proteome and nuclear proteome with RinID

As a demonstration of its broad applicability, we further extended RinID to profile the local proteomes at two other subcellular compartments: the ER membrane (ERM) and the nucleus. For the profiling of ERM proteome, HEK293T cells expressing translocon SEC61B-miniSOG fusion protein were labeled with 20 mM probe **6** for 15 min in four biological replicates, including two "+/− blue LED" replicates (with blue light omission as the negative control) and two "ERM vs. NES" replicates (with cytoplasmic targeted miniSOG as the control) (Supplementary Fig. 6A–B). A total of 943 and 532 proteins were identified and quantified in both replicates for "+/− blue LED" and "ERM vs. NES" datasets, respectively. For each dataset, proteins were ranked by their averaged H/L ratios and the cut-off ratios were determined with ROC analysis (Supplementary Fig. 6C, Supplementary Data 3). The 'true positive' and 'false positive' lists were both derived from the lists used

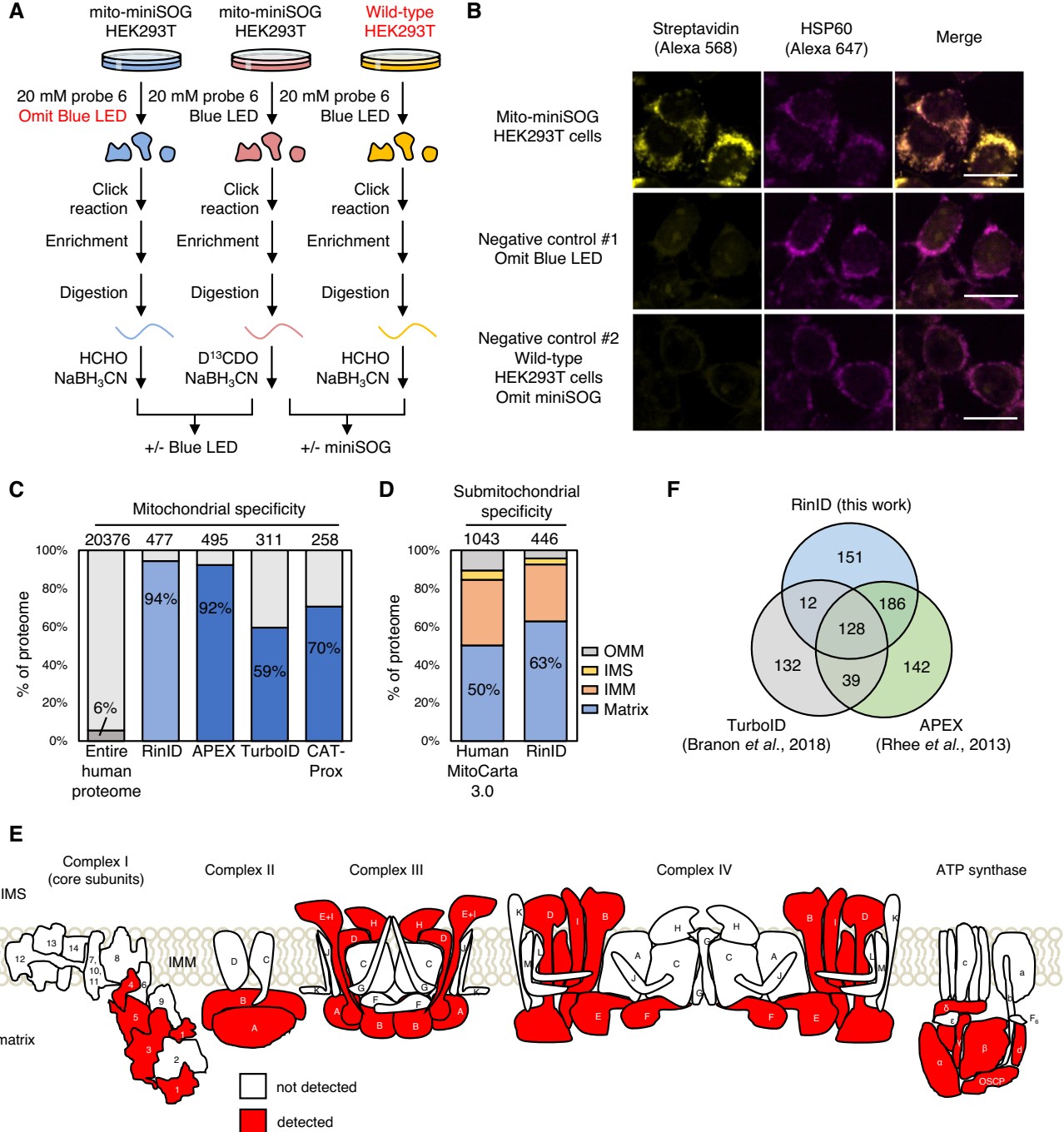

**Fig. 3 | Analysis of mitochondrial proteome identified by RinID. A** Schematic workflow of mass spectrometry-based proteomic analysis of RinID at mitochondrial matrix. **B** Confocal fluorescence images of HEK293T cells labeled with RinID and control samples omitting blue light or miniSOG. Scale bars: 20 μm. **C** Comparison of spatial specificity of RinID proteomic data with three other proximity labeling methods at mitochondria, Mitocarta 3.0 are defined as mitochondrial proteome. **D** Sub-mitochondrial specificity analysis for mitochondrial matrix RinID proteomic dataset. **E** Cartoon representation of RinID protein coverage in the electron transport chain complexes embedded in the mitochondrial inner membrane. miniSOG is targeted to the mitochondrial matrix side. **F** Comparison of mitochondrial proteome covered by RinID, APEX, and TurboID.

in previous ERM TurboID work[8], which consisted of 11838 secretory pathway annotated proteins and 7421 non-secretory pathway annotated proteins, respectively (Supplementary Data 3). The cut-off $\log_2 H/L$ ratios were set at 1.67 and −0.83, yielding 413 and 322 enriched proteins for "+/− blue LED" and "ERM vs. NES" datasets, respectively. The overlap of these two lists contained 150 proteins, which were defined as our ERM proteome (Supplementary Data 3).

Among the 150 ERM proteins, 93% (139 out of 150) are annotated as secretory pathway proteins, which is substantially higher than

specificity when compared to TurboID (72%)[8] and APEX2 (70%)[10] (Fig. 4A, B). In comparison, only 51% of the human proteome are secretory pathway proteins. Admittedly, the coverage of RinID datasets in both compartments are lower that previously reported TurboID and APEX2 datasets, suggesting that the labeling sensitivity needs to be further improved. Nevertheless, that RinID is capable of identifying proteins not previously covered by TurboID and APEX2 indicates that this new method could complement existing techniques by expanding the overall coverage.

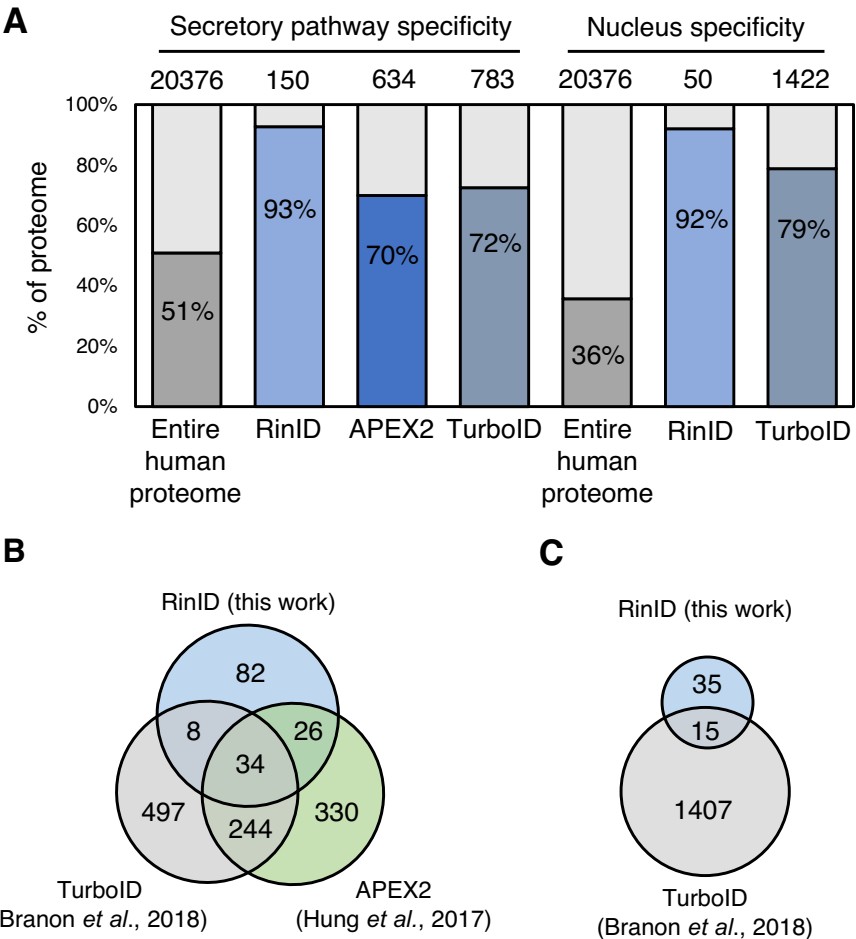

**Fig. 4 | Subcellular proteomic profiling with RinID in the ER membrane (ERM) and nucleus. A** Comparison of spatial specificity of proteomic data derived from RinID and other proximity labeling methods at ERM and nucleus. **B, C** Comparisons of proteomic coverage by RinID and other proximity labeling methods at the ERM (**B**) and nucleus (**C**). MiniSOG is fused with the histone protein H2B, which is targeted to chromatin. TurboID is fused with the nuclear localization sequence (NLS) and is targeted to the nucleoplasm.

For the profiling of nuclear proteome, we tested whether we could use lower concentration of probe **6** for MS identification. HEK293T cells expressing histone protein H2B-fused miniSOG were labeled with 10 mM probe **6** for 15 min in four replicated experiments, including two "+/− blue LED" replicates and two "H2B vs. NES" replicates (Supplementary Fig. 7A–C). We used the same 'true positive' and 'false positive' lists from previously published TurboID work[8], which contained 6710 nuclear annotated proteins and 6815 non-nuclear annotated proteins, respctively (Supplementary Data 4). Following a similar data analysis pipeline as the ERM experiments, a total of 119 and 119 proteins were enriched for "+/− blue LED" and "H2B vs. NES" datasets, respectively (Supplementary Fig. 7D, Supplementary Data 4), yielding an overlap of 50 proteins as our nuclear proteome (Supplementary Data 4). Gene Ontology analysis reveals that 92% (46 out of 50) of proteins in the list have prior nuclear annotations, which is higher than the specificity of TurboID nuclear proteome (79%)[8] and substantially higher than the percentage in the human proteome (36%) (Fig. 4A). The lower coverage of RinID may, in part, result from the use of histone protein H2B as the bait for proximity labeling (Fig. 4C). In contrast, TurboID was targeted to the nucleoplasm via fusion with a nuclear localization sequence. Thus, the proteome identified by RinID is more focused on histone and chromatin-associated proteins, but contains less other nuclear proteins. For example, among the top 9 out of 50 (18%) nuclear proteins enriched by H2B RinID, 8 (89%) have chromosome, chromatin, and/or DNA annotations in Gene Ontology. In contrast, among the top 18% nuclear proteins enriched by NLS TurboID,

only 54% have these annotations. The coverage of RinID nuclear proteome is higher than a recently reported proximity labeling method using chromatin-targeted small-molecule photocatalyst dibromofluorescein-Hoechst, in which only 10 nuclear proteins were identified[16].

### Pulse-chase RinID reveals differential protein retention in the ER lumen

The strong dependence of RinID on light illumination could be leveraged to achieve pulse-chase protein labeling (Fig. 5A). While long-term and intense irradiation of miniSOG could lead to excessive protein oxidation and even cell death[33], it might be possible to balance between efficient protein labeling and low cytotoxicity by carefully tuning the dosage of blue light. To reduce illumination time, we chose an engineered miniSOG variant, SOPP3, with improved singlet oxygen quantum yield[34]. We constructed Hela cell lines stably expressing SOPP3 targeted to the ER lumen. Western blot and immunofluorescence imaging analysis revealed stronger labeling intensity by SOPP3 over miniSOG with 5 min blue light illumination and 5 mM probe **6** (Supplementary Fig. 8A, C). Thereafter, labeled Hela cells were "chased" by culturing in the absence of probe **6** and blue light for 8 hours. Mitochondrial activity assay showed no changes in cell viability during the chase period (Supplementary Fig. 8B). These results indicate that pulse labeling with SOPP3 causes minimal cytotoxicity and is thus applicable for pulse-chase experimental schemes.

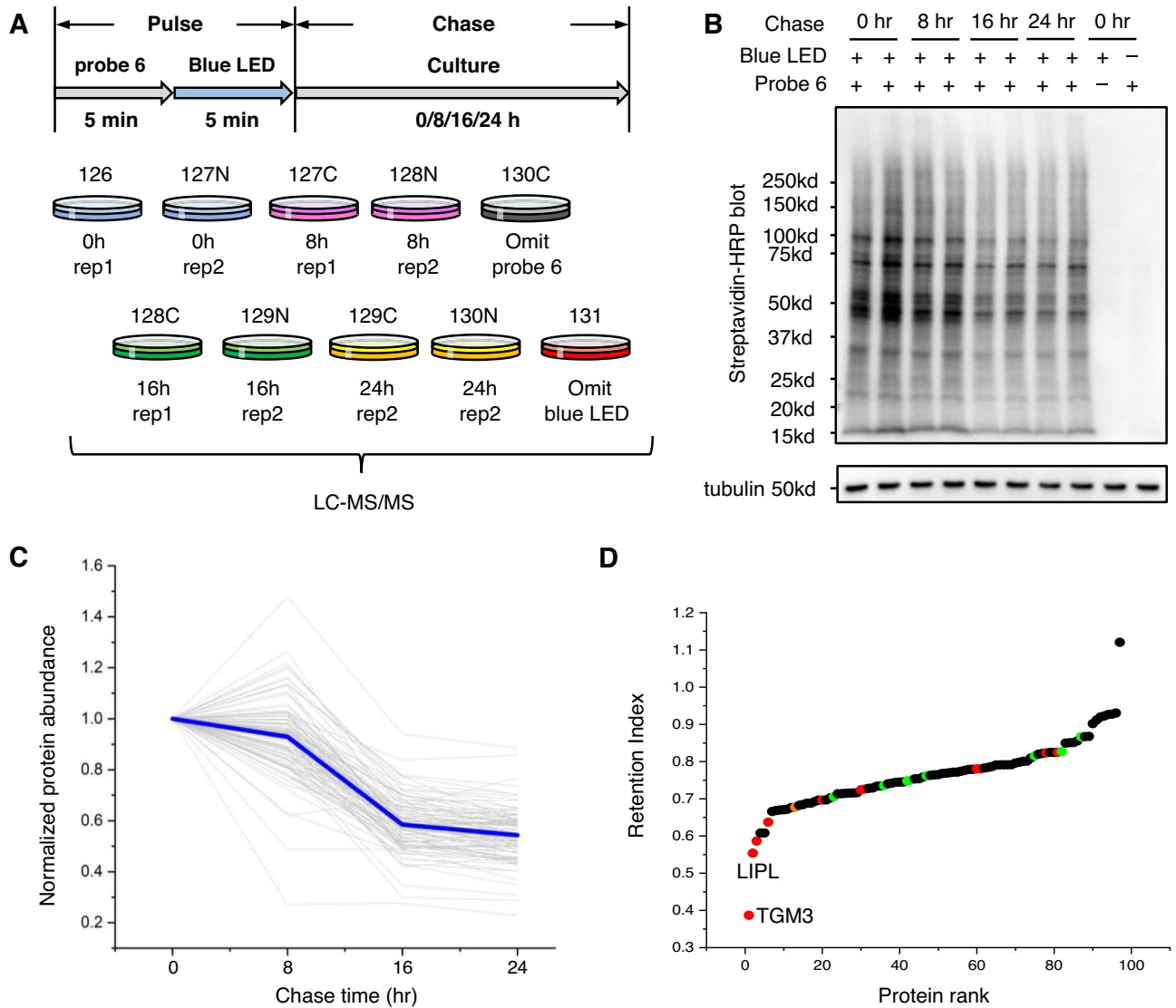

**Fig. 5 | Pulse-chase labeling of secretory pathway proteome with RinID.**
**A** scheme of pulse-chase RinID labeling and TMT-based quantitative proteomics.
**B** Western blot of the labeling signal at different chasing time point. SDS-PAGE
concentration: 4–20%. **C** Decrease ratio of RinID identified ER lumen proteins at
different chasing time point compared to 0 h chasing. Blue line indicates the
average of the identified proteins. **D** Retention index of ER lumen proteins identi-
fied in pulse-chase RinID. Red: secreted proteins; Green: ER-resident proteins;
Orange: ER resident with secreted subcellular location annotation.

We thus design a pulse-chase labeling scheme in Hela cells stably
expressing ER lumen targeted SOPP3 to monitor the clearance rate of
secretory pathway proteins by TMT labeling-based quantitative pro-
teomics. Cells were labeled with 5 mM probe **6** and blue LED illumi-
nation at 30 mW·cm$^{-2}$ for 5 min, which was followed by chasing in
normal cell culture medium for up to 24 h. Cells were sampled at 8 h
intervals, clicked with biotin-azide, and enriched with streptavidin-
coated agarose beads. Western blot and silver staining indicated that
the labeling signal decreased dramatically after 16 h chasing (Fig. 5B,
Supplementary Fig. 8D). Enriched proteins were digested by trypsin
and labeled with TMT reagents for isobaric quantitative LC-MS/MS
analysis (Fig. 5A). To account for the slight variations in sample load-
ing, we normalized the TMT reporter ion intensity of each protein with
respect to the signal of an endogenously biotinylated protein, pyr-
uvate carboxylase.

To determine the ER lumen proteome, we calculated the ratios of
the averaged reporter ion intensity of two replicates of 0 h sample (126
and 127 N) over the negative controls omitting either the probe (130 C)
or the blue light illumination (131). For ROC analysis, we used the same
'true positive' and 'false positive' lists as those in the previous ERM

analysis (Supplementary Data 5), which revealed cut-off log$_2$ ratios as
2.44 and 1.44 (Supplementary Fig. 8E) for +/− probe and +/− light,
respectively. We further filtered out proteins with averaged intensities
<10,000 in the 0 h samples (126 and 127 N), yielding a final list of 100
proteins. Notably, 97 out of 100 have secretory pathway annotations in
Gene Ontology, indicating high spatial specificity of RinID in the ER
lumen (Supplementary Fig. 8F, Supplementary Data 5).

Protein abundance at each time point in the chase period was
normalized with respect to the initial state of 0 h. We calculated the
ratios of the averaged reporter ion intensity of each protein at 8, 16,
and 24 h over those at 0 h (Fig. 5C, Supplementary Data 5). As expec-
ted, an overall downward trend was observed for protein abundance as
a function of chase time (A-t curve), with an averaged decrease of
7.0 ± 15.6%, 41.5 ± 10.5% and 45.7 ± 10.6% at 8, 16 and 24 h, respectively.
It is worth noting that some proteins showed increased signal at 8 h. To
test whether protein labeling could continue to occur after the blue
light irradiation is switched off, we designed the following experiments
comparing the protein labeling intensities in SOPP3-KDEL HeLa stable
cell line: (1) irradiating cells with blue LED in the presence of probe 6
for 5 min (normal labeling); (2) irradiating cells with blue LED in the

absence of probe 6 for 5 min, followed by incubating cells with probe 6 in the dark for 5 min (staggered labeling); (3–4) negative controls where cells are kept in the dark for 5 min, in either the presence (3: omit BL) or the absence (4: omit BL and probe) of probe 6. Streptavidin blot analysis of protein biotinylation confirms that the staggered labeling sample (2) exhibits minimal labeling signal that is on par with the two negative controls (3–4), while the normal labeling sample (1) has substantially higher biotinylation signal (Supplementary Fig. 8G). This demonstrates that probe 6 would not react with the oxidized proteins after the blue LED is switched off. Thus, we attribute the increased retention index to the measurement errors introduced during MS quantitation. Within this decreasing background, there were considerable variations in the clearance rate among individual proteins. To quantitatively measure protein clearance, we defined Retention Index (RI) for each protein as the area under its A-t curve (Supplementary Data 5). Figure 5D ranks proteins according to their RI. Notably, a few proteins exhibit substantially lower RI, indicating faster clearance. Gene Ontology annotations reveal terms related to secretion for these proteins (Fig. 5D). Indeed, among the 12 secreted proteins in our dataset, 6 are rapidly cleared from cells with RI lower than 0.70. The fastest two proteins, TGM2 and LIPL, are already removed by 73 and 51% at 8 h. In contrast, ER resident proteins typically have higher RI, with an average of $0.79 \pm 0.06$ for the 9 proteins in our dataset. The observed variations in protein clearance dynamics attests to the complexity of the mechanisms by which cells regulate proteomic homeostasis. Through the above proof-of-concept experiment, we demonstrate the feasibility of monitoring subcellular proteomic dyanamics with RinID, which is the first proximity labeling method that is compatible with pulse-chase labeling scheme.

## Discussion

In summary, we have developed RinID, a light-activated proximity-dependent protein labeling method for profiling subcellular proteome. We characterized its labeling mechanism and identified amino acid residue of photo-oxidation. Using two cell lines and multiple subcellular organelles (i.e., mitochondria, nucleus, and ER), we demonstrated the exceptional spatial specificity of RinID by both fluorescence imaging and quantitative proteomics. We also reported the first case of pulse-chase proximity labeling with RinID for monitoring protein clearance in the secretory pathway of HeLa cells.

In the pulse-chase RinID labeling, a critical question is whether the RinID labeling itself may affect the observed protein retention patterns. Although it is experimentally difficult to completely rule out such possibility, we note that the power and the time window of blue light irradiation used in pulse-chase RinID (30 mW·cm$^{-2}$ LED, 5 min) are substantially smaller than those used in CALI[35] (540 mW·cm$^{-2}$ laser, 5 min; or 70 mW·cm$^{-2}$ laser, 25 min). Indeed, cell viability was not significantly changed at 8 hours post-RinID labeling (Supplementary Fig. 8B), and most of the proteins identified by pulse-chase RinID did not exhibit high degradation rate (Fig. 5C, D). The above evidence suggests that protein labeling by SOPP3 may not substantially affect the retention pattern of the identified proteins during the chase period. However, users of RinID should be cautious about the potential perturbation effect, particularly when using cells that are sensitive to oxidative stress.

Over the past decade, engineered peroxidases (e.g., APEX[5], APEX2[6], HRP[36]) and biotin ligases (e.g., BioID[7], TurboID[8]) have been employed as powerful tools for proximity labeling. The spatial specificity and coverage of RinID is comparable to these methods. Meanwhile, the difference in amino acid preference suggests that RinID may complement existing techniques to improve the overall coverage. Compared to APEX, RinID is less toxic by avoiding the use of $H_2O_2$. Compared to TurboID, the light-triggered RinID labeling offers better temporal control of the reaction, which is more suitable for studying dynamic changes in the subcellular proteome. Compared to small-molecule photocatalysts, genetically encoded RinID could be more readily targeted to subcellular locations.

Recently, a method termed photoactivation-dependent proximity labeling (PDPL) that also applied miniSOG for subcellular proteome profiling has been reported[37]. Although both RinID and PDPL used miniSOG as the photosensitizer, the probe and the time of probe incubation and blue light irradiation are different in the two methods. RinID used higher concentration of primary amine probe to gain a shorter probe incubation and blue light irradiation time. The quantitative proteomics methods are also different in the two methods. Due to the above differences, RinID gained a higher subcellular specificity (more than 90%, PDPL is <80%) in all of the three subcellular structures, while PDPL showed a higher coverage. More importantly, the PDPL work focused on the protein–protein interaction identification, while RinID focused on the pulse-chase proximity labeling to monitor the subcellular protein dynamics. Both works have shown that genetically encoded photosensitizers are powerful tools for the proximity labeling of subcellular proteomes with high spatiotemporal resolution.

While RinID has proven efficient in multiple subcellular organelles, a few problems still need to be solved in the future. First, the background generated by probe **6** needs to be deducted by quantitative MS experimental design, which may complicate data analysis. Second, the 15 min labeling time is still longer than APEX2. Third, the tissue penetration of blue light is typically restricted to <1 mm, thus limiting applications to tissue samples such as brain slice. Similar to other proximity labeling methods, these problems could be solved by the development of better probes with higher specificity and labeling efficiency, and by the design of photocatalysts with higher quantum yield of singlet oxygen and red-shifted excitation spectrum. With further development of such probes and photocatalysts, RinID would become a powerful tool for high spatiotemporal resolution identification of subcellular proteomes.

## Methods
### Reagents
All the information about the reagents, antibodies, and plasmids used in this work can be found in Supplementary Tables 1–3, separately. Chemical probe synthesis is described in the Supplementary Methods.

### Buffer recipes

- RIPA lysis buffer: 25 mM Tris·HCl pH 7.6, 150 mM NaCl, 2% SDS, 1% sodium deoxycholate, 1% NP-40
- LB culture medium: 10 g NaCl, 10 g tryptone, 5 g yeast extract in 1 L ddH$_2$O
- Binding buffer: 50 mM Tris·HCl pH 7.6, 0.3 M NaCl
- Bacteria lysis buffer: binding buffer with 1× proteinase inhibitor cocktail
- Washing buffer: binding buffer with 30 mM imidazole
- Elution buffer: binding buffer with 200 mM imidazole
- 10x SDS-PAGE Running buffer: 76 g Tris base, 360 g glycine, 25 g SDS in 2.5 L ddH$_2$O
- 10x SDS-PAGE Transfer buffer: 76 g Tris base, 360 g glycine in 2.5 L ddH$_2$O
- Urea buffer: 3.844 g urea and 800 uL 50 mM Tris·HCl (pH 8.5) in 7.2 mL ddH$_2$O
- PBST: 0.2% Tween-20 in PBS

### Mammalian cell culture
Wild-type HEK293T cells were purchased from American Type Culture Collection (ATCC); Wild type HeLa cells were purchased from the National Biomedical Experimental Cell Resource Library of China (BMCR) (Resource number: 1101HUM-PUMC000011). HEK293T cells or Hela cells were maintained in DMEM (Gibco, C11995500BT)

supplemented with 10% FBS (Gibco, 10099141 C). Cells were cultured at 37 °C under 5% $CO_2$.

For lentivirus production, HEK293T cells cultured in six-well plates at ~60% confluence were co-transfected with the gene of interest in a lentiviral vector pLX304 (1 µg) and two packaging plasmids, dR8.91 (1 µg) and pVSV-G (700 ng), mixed with 10 µL PEI in 1 mL DMEM for 4 h. Then the cells were further cultured in DMEM with 10% FBS for 48 h. Then the culture medium containing lentivirus was collected and filtered through a 0.45 µm filter. Thereafter, 1 mL of the lentivirus-containing medium was added to fresh HEK293T cells at ~70% confluence in a well of a six-well plate (~600,000 cells). 48 h after lentiviral transduction, the culture medium was exchanged to fresh complete medium containing 5 µg·mL$^{-1}$ blasticidin (Selleck, S7419) for selection. Infected cells were maintained in the selection medium for 7 days, with daily change of fresh selection medium. miniSOG expression in selected cells were verified via immunofluorescence. These cell lines were maintained in 5 µg·mL$^{-1}$ culture medium supplemented with blasticidin.

### Protein expression and purification
BL21 competent bacteria were transformed with pET21a-miniSOG-6×Histag plasmid (or pET21a-sortaseA-6×Histag plasmid) and cultured in 500 mL LB culture medium at 37 °C for about 12 h until the OD reached 0.8. Following the addition of 0.5 mM IPTG (final concentration) to the medium, bacteria were cultured at 18°C for another 14 h to allow protein expression. The 500 mL bacterial culture was centrifuged at 4000 × $g$ for 10 min at 18 °C. The supernatant was discarded and the pellet was re-suspended in 20 mL bacteria lysis buffer. Following sonication on ice for 45 min, the bacterial cell lysate was centrifuged at 17,000 × $g$ for 20 min at 4 °C. The supernatant was filtered through a 0.22 µm filter, and the filtrate was stored in a 50 mL centrifuge tube.

For affinity purification, 5 mL Ni-NTA beads (Qiagen, 30210) was washed with 10 mL binding buffer in a column and then mixed with the protein filtrate for 30 min under gently rotation. The flow-through was collected and labeled as 'Flow 1'. Thereafter, 10 mL binding buffer was added to wash off non-specifically adsorbed proteins. The flow-through was collected and labeled as 'Flow 2'. Then 10 mL wash buffer was added to wash off proteins that have weak interactions with the Ni-NTA column and the eluent was collected as 'Wash'. Finally, 15 mL of elution buffer was added to the column, and eluent was collected into 1.5 mL centrifuge tubes with 1 mL per fraction.

Protein samples were analyzed with SDS-PAGE. The eluted samples with three highest concentrations were combined and purified by ultrafiltration and then combined. The concentration of purified protein was measured by NanoDrop 2000C (Thermo). The purified protein was then stored at −80 °C.

### Gel electrophoresis and western blot analysis
For SDS-PAGE analysis, 10 µL of each eluted fraction was sampled and mixed with 2.5 µL 5x protein loading buffer. Following sample boiling at 95 °C for 10 min, each sample was loaded to a 12% or 4–20% gradient SDS-PAGE gel and analyzed by electrophoresis. The gel was stained with Coomassie brilliant blue (CBB) and imaged on a ChemiDoc MP Imaging System (Bio-Rad).

For western blot analysis, the protein gel was transferred to a PVDF membrane (Bio-Rad) under 230 mA for 1 h. The membrane was blocked with blocking buffer (5% BSA in TBST) at room temperature for 30 min and then immersed with 0.2–0.4 µg/mL streptavidin-HRP in TBST at room temperature for 1 h. For anti-V5 or anti-tubulin western blot, the membrane was incubated with mouse anti-V5/anti-tubulin primary antibody (1:5000 dilution) overnight, followed by anti-mouse secondary antibody conjugated with HRP (1:5000 dilution) for 1 h. Antibodies used in this study can be found in Supplementary Table 2. The membrane was washed by TBST three times after each step of

incubation. The blots were developed with Clarity Western ECL substrate (Bio-Rad) and imaged by ChemiDoc MP Imaging System.

### miniSOG-mediated labeling of purified proteins and cell lysate
For biotinylation, 1 mM BSA, 20 mM Biotin-PEG-NH$_2$ (Biomatrik, 246702, probe 1) or probe 6 and 100 µM miniSOG in PBS (Solarbio, pH 7.2–7.4, P1020-500mL) (or 1 mg/mL sortase A, 10 mM Biotin-PEG-NH$_2$, and 100 µM miniSOG) were mixed in 50 µL reaction volume and then illuminated with 19 mW·cm$^{-2}$ blue light-emitting diode (LED) at room temperature for 30 min. Labeled BSA was desalted by Bio-Rad Micro Bio-Spin P-30 Gel column, and then reacted with 100 µM N$_3$-biotin (Aldrich, 762024) in the presence of 667 µM CuSO$_4$, 1.3 mM BTTAA and 2.5 mM sodium ascorbate. The reaction mixture was incubated at room temperature for 60 min. Samples omitting blue light, miniSOG, probe, and/or click reaction were used for negative controls. The reaction mixture was diluted to 2.50 mL with PBS and analyzed with 12% or 4–20% gradient SDS-PAGE and western blot.

For the MS sample preparation, 50 µL of labeled BSA sample was incubated with 400 µL cold methanol at −80 °C to precipitate the protein. The precipitate was dissolved by 660 µL urea buffer. 0.5 M DTT was added to reach a final concentration of 10 mM and the sample was incubated at 55 °C for 25 min. Then, 0.5 M IAA was added to reach a final concentration of 30 mM and the sample was incubated at 37 °C for 30 min in dark. 0.5 M DTT was added again to reach a final concentration of 20 mM and the sample was incubated at 37 °C for 15 min in dark. 7 volumes of 50 mM Tris-HCl (pH 8.5) was added. 100 mM CaCl$_2$ was added to reach a final concentration of 1 mM. 1 mL of the sample containing 77 µg BSA was digested with 3.85 µg trypsin (Sigma) at 37 °C for 16 h. The digested peptides was dried by rotary evaporator (1500 × $g$, 45 °C, 4–6 h), desalted on a Pierce C18 tip (Thermo, 87784), and re-dried by rotary evaporator (1500 × $g$, 45 °C, 4–6 h). The two desalted peptide samples (BL+ and BL−) were identified by LC-MS/MS as described in the section 'Liquid chromatography-tandem mass spectrometry'.

For cell lysate labeling, HEK293T cells expressing miniSOG were cultured in a 10-cm dish and harvested into 700 µL PBS containing protease inhibitor. The cell suspension was lysed with sonication and dispensed into 6 tubes, with 100 µL in each tube. Each of the probes (10 mM probe 1 and probe 6, 1 mM for probes 2−5) was added into a tube. Then each 100 µL mixture was divided into two halves. One sample was mixed with 1 mg/mL miniSOG and illuminated with 19.1 mW·cm$^{-2}$ blue LED for 15 min. The other sample was the negative control. The samples were analyzed with 4-20% gradient SDS-PAGE and western blot analysis.

### RinID labeling in live cells
HEK293T cells or Hela cells stably expressing miniSOG or SOPP3 were cultured in 6-well plates. After reaching 90% confluence, the cell culture media were discarded and the cells were washed by 1×PBS once, and then incubated with the probe at the indicated concentration (10 mM for probe 1; 0.5 mM for probes 2−5; and 5, 10, 20 mM for probe 6) in pre-warmed Hanks Balanced Salt Solution (HBSS, Solarbio, H1025) at 37 °C for the indicated period of time (30 min for probes 1−5; 5 min for probe 6). Labeling was triggered by blue LED irradiation at room temperature. Then the cells were scraped and collected by centrifugation at 300 × $g$ and 4 °C for 5 min. The pellet was lysed on ice in 200 µL RIPA lysis buffer containing 1×protease inhibitor (Roche, 4693159001) by sonication. The cell lysate was centrifuged at 15,000 × $g$ at 4 °C for 10 min. 100 µL supernatant was collected and 800 µL cold methanol was added to precipitate proteins at −80 °C overnight. The samples were then centrifuged at 8000 × $g$ for 5 min at 4 °C. The methanol was discarded and the precipitate was re-dissolved in 100 µL 0.5% SDS aqueous solution. The sample was mixed with 50 µL click cocktail (final concentration: 100 µM N$_3$-biotin, 667 µM CuSO$_4$, 1.3 mM BTTAA and 2.5 mM sodium ascorbate) or PBS

(negative control) and reacted at room temperature for 60 min. The biotinylated protein samples were analyzed with 12% or 4–20% SDS-PAGE and western blot.

The optimized conditions for miniSOG-mediated labeling were: 0.5–5 mM probe **2** for 15 min illumination and 20 mM probe **6** for 15 min illumination (Fig. 2C, Supplementary Fig. 2C). 10 and 20 mM PA were used for the preparation of MS sample for nuclear labeling and mitochondria matrix/ERM labeling, respectively. The optimized conditions for SOPP3-mediated labeling were: 5 mM probe **6** for 15 min illumination. This condition was also used for pulse-chase labeling in Hela cells.

### Fluorescence microscopy

Cells were cultured on glass coverslips in 24-well plates at a density of ~70,000 cells per well. To improve the adherence of HEK293T cells, glass coverslips were pretreated with 20% Corning Matrigel matrix (Corning, 356234) diluted in DMEM overnight at 37 °C and washed with PBS once before use. After 48 h, cells were washed with PBS once, incubated with 20 mM probe **6** in HBSS for 5 min at 37 °C and then illuminated with 30 mW·cm$^{-2}$ blue LED for 15 min at room temperature.

Thereafter, cells were washed with PBS once and fixed with cold methanol (−20 °C) (caution: do not use formaldehyde to fix cells to avoid the crosslink between PA probe and protein/nuclear acids that could increase the background signal) at −20 °C for 15 min and then washed with PBS three times. Then 300 µL of click reaction reagents were added to each well (final concentration: 100 µM N$_3$-biotin, 667 µM CuSO$_4$, 1.3 mM BTTAA, and 2.5 mM sodium ascorbate) and incubated at room temperature for 30 min. Then cells were washed with PBS three times and then blocked with 3% BSA in PBST for 30 min at room temperature.

For immunostaining, cells were incubated with primary antibody (mouse anti-V5 or anti-HA antibody at 1:1000 dilution, and rabbit antibody of organelle markers at the recommended dilution ratio by the supplier) for 1 h at room temperature. After washed with PBST three times, cells were incubated with DAPI (ThermoFisher, D1306, 1:5000 dilution) as well as secondary antibody (goat anti-mouse Alexa Fluor 647 and anti-rabbit Alexa Fluor 488 at 1:1000 dilution) and streptavidin-Alexa Fluor 568 (ThermoFisher, S21374, 1:1000 dilution) for 1 h at room temperature. Cells were then washed three times with PBST at room temperature. Cells were maintained in PBS for imaging at room temperature (20 °C). Antibodies used in this study can be found in Supplementary Table 2.

Immunofluorescence images were collected with an inverted fluorescence microscope (NikonTiE) equipped with a 40× oil immersion objective lens (NA 1.3), four laser lines (Coherent OBIS, 405, 488, 561, and 637 nm), a spinning disk confocal unit (Yokogawa CSU-X1), and a scientific CMOS camera (Hamamatsu ORCA-Flash 4.0 v2). The overall imaging equipment was controlled with a customized software written in LabVIEW (National Instruments). All images were analyzed by ImageJ software. The scale of confocal images is 1500 ×1000 pixel, 6.5 µm for each pixel. The images shown in Fig. 2E are 268 × 243 pixel, in Supplementary Figs. 4 and 8C are 1500 × 1000 pixel. Scale bars indicate the real length represented by the images.

### Mass spectrometry sample preparation

HEK293T cells stably expressing mito-V5-miniSOG (or V5-miniSOG-sec61b, or V5-H2B-miniSOG) or Hela cells stably expressing ss-HA-SOPP3-KDEL were cultured into 10 cm dishes. After reaching 90% confluence, DMEM and 10% FBS was discarded and cells were incubated with 20 mM probe **6** (10 mM for V5-H2B-miniSOG, 5 mM for SOPP3-KDEL Hela) in HBSS for 5 min at 37 °C. Labeling was triggered by 30 mW·cm$^{-2}$ blue light for 15 min (5 min for SOPP3-KDEL Hela) at room temperature. Then the cells were scraped and collected by centrifugation at 300 × g and 4 °C for 5 min. The pellet was lysed on ice using 600 µL RIPA lysis buffer for 15 min.

For pulse-chase labeling in SOPP3-KDEL Hela cells, labeled cells were washed by 1 mL PBS for three times and then re-cultured in DMEM and 10% FBS at 37 °C for various durations (0, 8, 16, 24 hours). Cells were then washed two times by PBS, scraped, and collected from the dish by 600 µL RIPA and lysed on ice by sonication. The lysate was centrifuged at 15,000 × g at 4 °C for 10 min. Excessive small molecules were removed by protein precipitation in cold methanol at −80 °C overnight. The protein samples were centrifuged at 3000 × g at 4 °C for 10 min. The protein pellet was washed with cold methanol (−80 °C) twice and then dissolved by 600 µL 0.5% SDS aqueous solution. 300 µL of click reaction reagents were added to each tube (final concentration: 100 µM N$_3$-biotin, 667 µM CuSO$_4$, 1.3 mM BTTAA, and 2.5 mM sodium ascorbate) and incubated at room temperature for 1 h. Protein samples were analyzed by 12% or 4–20% gradient SDS-PAGE and western blot. The remaining samples were precipitated with cold methanol and stored at −80 °C overnight.

The protein pellet was washed with cold methanol (−80 °C) twice and then dissolved by 800 µL 0.5% SDS aqueous solution. The protein concentration was measured with BCA protein assay with Pierce$^{TM}$ BCA Protein Assay kit (Thermo, 23227) before enrichment. After adjusting the concentration, 40 µL of the protein sample was taken as 'input'. Thereafter, 50 µL streptavidin agarose resin (Thermo, 20347) was washed by 1 mL PBS buffer once, and was incubated with the protein solution at 25 °C for 3 h with gentle rotation.

The agarose beads were centrifuged at 3000 × g for 2 min. 40 µL of supernatant was taken as 'supernatant' before discarding the supernatant. Then beads were washed once with 1 mL 0.5% SDS in PBS for 10 min with gentle rotation and six times with 1 mL PBS successively. 40 µL of 0.5% SDS washed supernatant was taken as 'Wash 1'. Thereafter, the beads were centrifuged at 3000 × g for 2 min before discarding the supernatant, and then were resuspended in 50 µL PBS and taken 20 µL as 'elute'.

After discarding the supernatant, the beads were resuspended by 500 µL 6 M urea in PBS buffer. 25 µL 200 mM dithiothreitol (sigma, D9163-5G) aqueous solution was added and incubated at 60 °C for 15 min. The beads were cooled to 25 °C. Then, 25 µL 400 mM iodoacetamide (Sigma, I6125-5G) aqueous solution was added and incubated at 30 °C for 30 min in the dark. Beads were washed twice with 1 mL 100 mM triethylammonium bicarbonate buffer (Sigma, T7408-100mL) and resuspended in 200 µL triethylammonium bicarbonate buffer. 1 µg sequencing-grade trypsin (Promega, V5111) was added in each tube for the on-beads digestion by shaking at 1200 rpm and 37 °C for 16 h. Thereafter, released peptides were collected from the supernatant by centrifugation at 15,000 × g for 10 min, the pellet was discarded.

For miniSOG labeled samples, dimethylation labeling was taken for quantitative proteomics identification. Each peptide sample (200 mL) was mixed with 12 µL 4% (v/v) CH2O (Sigma, 252549-25 ml) or 12 µL 4% (v/v) $^{13}$CD$_2$O (Sigma, 596388-1g) respectively, and 12 µL 39.68 mg/mL NaBH$_3$CN (sigma, 156159-10 G) was added. The solution was incubated at room temperature for 1 h with 1200 rpm shaking. The reaction is stopped by adding 48 µL 1% (v/v) ammonia solution (Aladdin, A112079) and 24 µL formic acid (Fluka, 94318-50 ml), the light and heavy isotopically labeled samples were mixed and dried by rotary evaporator (1500 × g, 45 °C, 4–6 h) and stored at −80 °C.

For SOPP3 mediated pulse-chase labeled samples, TMT 10 plex Mass Tag Labeling Kits and Reagents (Thermo, 90110) was used for quantitative proteomics identification. Peptide samples were first dried by rotary evaporator (1500 × g, 45 °C, 4–6 h) and desalted by Pierce C18 tips 100 µL (Thermo, 87784). The desalted samples were re-dried as described before. Re-dried samples were labeled by different TMT reagents (0 h, 126 and 127 N; 8 h, 127 C and 128 N; 16 h, 128 C and 129 N; 24 h, 129 C and 130 N; omitting probe **6**, 130 C; omitting blue light, 131), then combined, dried and stored at −80 °C.

Thereafter, the peptide sample was fractionized by Pierce High pH Reverse Phase Peptide Fractionation Kit (Thermo, 84868). Then the samples were combined as '1 + 5', '2 + 6', '3 + 7' and '4 + 8'. Combined samples were dried again by rotary evaporator (1500 × g, 45 °C, 4–6 h), and then identified by liquid chromatography–tandem mass spectrometry (LC–MS/MS).

## SDS-PAGE and silver staining

10 µL 5× protein loading buffer was added to 40 µL the protein sample mentioned in previous step named 'input', 'supernatant' and 'wash 1', 1 µL 50 mM Biotin and 5 µL 5× protein loading buffer was added to 20 µL the protein sample mentioned in previous step named 'Elute'. Then the samples were boiled at 95 °C for 10 min. Use 1× protein loading buffer to dilute the samples to make sure the final loading ratio of 'input', 'supernatant', and 'wash1' is 1/100 of elute (e.g., 'input' is 10 µL from 800 µL, thus is 1.25% of the total volume, 'elute' is 10 µL from 200 µL, thus is 5% of the total volume, so 'input' should be further diluted by 1× protein loading buffer for 25 times to become 0.05% of the total volume, which is 1% of that of 'elute'). After adjusting the concentration, the labeled protein was separated by 12% or 4–20% SDS-PAGE gel, then analyzed by Fast Silver Stain Kit (Beyotime, P0017S) and imaged by ChemiDoc MP Imaging System (Bio-Rad).

## Liquid chromatography-tandem mass spectrometry

Peptides were separated using a loading column (100 µm × 2 cm) and a C18 separating capillary column (75 µm × 15 cm) packed in-house with Luna 3 µm C18(2) bulk packing material (Phenomenex, USA). The mobile phases (A: water with 0.1% formic acid and B: 80% acetonitrile with 0.1% formic acid) were driven and controlled by a Dionex Ultimate 3000 RPLC nano system (Thermo Fisher Scientific). The LC gradient was held at 2% for the first 8 min of the analysis, followed by an increase from 2 to 10% B from 8 to 9 min, an increase from 10 to 44% B from 9 to 123 min, and an increase from 44 to 99% B from 123 to 128 min.

For the samples analyzed by Orbitrap Fusion LUMOS Tribrid Mass Spectrometer, the precursors were ionized using an EASY-Spray ionization source (Thermo Fisher Scientific) source held at +2.0 kV compared to ground, and the inlet capillary temperature was held at 320 °C. Survey scans of peptide precursors were collected in the Orbitrap from 350 to 1600 Th with an AGC target of 400,000, a maximum injection time of 50 ms, RF lens at 30%, and a resolution of 60,000 at 200 $m/z$. Monoisotopic precursor selection was enabled for peptide isotopic distributions, precursors of $z = 2–7$ were selected for data-dependent MS/MS scans (with a resolution of 15,000) for 3 s of cycle time, and dynamic exclusion was set to 15 s with a ±10 ppm window set around the precursor mono-isotope.

In HCD scans, an automated scan range determination was enabled. An isolation window of 1.6 Th was used to select precursor ions with the quadrupole. Product ions were collected in the Orbitrap with the first mass of 110 Th, an AGC target of 50,000, a maximum injection time of 30 ms, HCD collision energy at 30%, and a resolution of 15,000.

## Mass spectrometry data analysis

The MS data of BSA samples were analyzed with MaxQuant v1.6.10 software. The quantification of light/heavy ratios was calculated with precursor mass tolerance of 20 ppm. For protein ID identification, MS/MS spectra were searched against UP000009136 proteome database from Uniprot (37,510 bovine proteins in total). Half-tryptic termini and up to 1 missing trypsin cleavages were allowed. Carbamidomethylation at cysteine (+57.0215 Da), oxidation at histidine, tyrosine, tryptophan, and methionine residues (termed 'H-oxidized' (+31.9898 Da), 'Y-oxidized' (+15.9949 Da), 'W-oxidized' (+3.9949 Da), 'M-oxidized' (+31.9898 Da),)), and the product of the reaction between probe **6** and the oxidation product of histidine, tyrosine, tryptophan, and methionine residue (termed 'H-PA'(+69.0215 Da), 'Y-PA' (+71.0371 Da), 'W-PA' (+69.0215 Da), 'M-PA' (+53.0265 Da)) were set as fixed modifications (structure of the proposed products are shown in Supplementary Fig. 1). Oxidation at methionine (+15.9949 Da) and acetylation of N-terminal (+42.0106 Da) were set as variable modifications.

Peptides of BSA (Uniprot ID: P02769) contain 'H-oxidized' and 'H-PA' sites were inspected. The peptide 'FKDLGEEHFK' that contains H18 residue of BSA, together with the peptide 'HLVDEPQNLIK' and 'LKHLVDEPQNLIK' that contain H378 of BSA were identified with both of the H-oxidized and H-PA only in the labeled sample. The MS/MS spectrum of 'FKDLGEEHFK' with 'H-oxidized' (scan number 4360, mass 1280.604) or 'H-PA' (scan number 4316, mass 1317.635) were analyzed and shown in Fig. 1C.

For cellular labeling samples, each biological replicate was pre-fractionated on a Pierce® High pH Reverse Phase Peptide Fractionation kit (Thermo, 84868). For mito-miniSOG labeling, two biological replicates were performed with -LED as the negative control, while another two were performed with the wild type HEK293T cells as the negative control. For miniSOG-sec61b and H2B-miniSOG labeling, two biological replicates were performed with -LED as the negative control, while another two were performed with miniSOG-NES as the negative control. For SOPP3-KDEL pulse-chase labeling, each time point has two replicates labeled by different TMT tags.

For protein identification, MS/MS spectra were searched against UP000005640 proteome database from Uniprot (79052 human proteins in total). Half-tryptic termini and up to 1 missing trypsin cleavages were allowed. Carbamidomethylation at cysteine (+57.0215 Da) and isotopic modifications (+28.0313 and +34.0631 Da for light and heavy labeling, respectively) at lysine/N-terminal were set as fixed modifications. Oxidation at methionine (+15.9949 Da) and acetylation of N-terminal (+42.0106 Da) were set as variable modifications. Each of the biological replicates was analyzed separately. Contaminants and proteins identified as reverse hits were removed. For miniSOG labeled samples, proteins with unique peptides <2 or H/L ratio 'NaN' were also removed. The quantification of light/heavy ratios was calculated with precursor mass tolerance of 20 ppm. For SOPP3 labeled samples, proteins with unique peptides <1 or any reporter ion intensity "0" were removed. The tolerance of the molecular weight of reporter ion on MS/MS is 0.003 Da.

For pulse-chase labeled ER lumen proteome, we used the ratios between reporter ion intensity of 0 h-chased samples (average of 126 and 127 N) and omitting probe **6** (130 C) or omitting blue light (131) as the enrichment fold. The 387 proteins identified were ranked by $\log_2$ ((126 + 127 N)/(2*130 C)) and $\log_2$ ((126 + 127 N)/(2*131)), respectively. 162 and 233 passed the cut-off with $\log_2$ ((126 + 127 N)/(2*130 C)) and $\log_2$ ((126 + 127 N)/ (2*131)) over 2.440 and 1.444. Besides, proteins with average of 126 and 127 N intensities <10,000 were also removed. The overlap of proteins above the three cut-offs yielded 100 proteins, with 97 annotated as secretory pathway proteins.

## Receiver operating characteristic (ROC) analysis

For all four proteomes identified, we performed ROC analysis to determine the cut off ratios. For mitochondria matrix proteome, 1132 proteins in human MitoCarta 3.0[2] were defined as 'true positive', 2403 proteins used as non-mitochondria protein in TurboID work[3] were defined as 'true negative'. For ERM/ER lumen proteome, 11,838 proteins with secretory annotation and 7421 proteins with non-secretory annotation in TurboID work were defined as 'true positive' and 'true negative', respectively. For nuclear proteome, 6710 proteins with nuclear annotation and 6815 proteins with non-nuclear annotation in TurboID work were defined as 'true positive' and 'true negative', respectively.

Mitochondria proteome. For the data derived from the two replicates with omitting blue LED irradiation or miniSOG as negative controls, the proteins are ranked by their averaged H/L ratio and taken for the ROC analysis. Of the 1634 and 1882 proteins, 611 and 556 of them passed the cut off with $\log_2$ H/L ratio over 1.213 and 1.245 (Supplementary Fig. 5C). The overlap of them gave 477 proteins (450 in human mitocarta 3.0, 94% mitochondria specificity), which is the mitochondria proteome identified by RinID. The 495, 311, and 258 proteins identified by APEX, TurboID, and CAT-Prox are compared with human mitocarta 3.0 and giving 92%, 59%, and 70% mitochondria specificity, respectively.

ERM proteome. We use search the Uniprot GOCC terms containing the following words to define 'secretory pathway protein': 'endoplasmic reticulum', 'Golgi', 'plasma membrane', 'extracellular', 'endosom', 'lysosom', 'nuclear envelope', 'nuclear membrane', 'perinuclear region of cytoplasm', and 'vesicle'. For the data derived from the two replicates with omitting blue LED irradiation or miniSOG as negative controls, the proteins are ranked by their averaged H/L ratio and taken for the ROC analysis. Of the 943 and 532 proteins, 413 and 322 of them passed the cut off with log2 H/L ratio over 1.667 and −0.831 (Supplementary Fig. 6C). The overlap of them gave 150 proteins (139 are secretory pathway proteins, 93% secretory pathway specificity), which is the ERM proteome identified by RinID. The 783 and 634 proteins identified by TurboID and APEX2 give 72 and 70% secretory pathway specificity, respectively.

Nuclear proteome. We use the following 7 GO terms to define 'nuclear protein': GO:0016604, GO:0031965, GO:0016607, GO:0005730, GO:0001650, GO:0005654, GO:0005634. For the data derived from the two replicates with omitting blue LED irradiation or miniSOG as negative controls, the proteins are ranked by their averaged H/L ratio and taken for the ROC analysis. Of the 301 and 239 proteins, 119 and 119 of them passed the cut off with log2 H/L ratio over 0.820 and −0.644 (Supplementary Fig. 7D). The overlap of them gave 50 proteins (46 are secretory pathway proteins, 92% nuclear specificity), which is the nuclear proteome identified by RinID. The 1442 proteins identified by TurboID give 79% nuclear specificity.

## Cell viability assay
HeLa cells stably expressing SOPP3-KDEL were cultured into 96-well plates. After reaching 90% confluence, DMEM and 10% FBS was discarded and cells were incubated with 5 mM probe 6 in HBSS for 5 min at 37°C. Labeling was triggered by 30 mW·cm$^{-2}$ blue light for 5 min at room temperature. Then the cells were washed by 100 μL PBS for three times and cultured in DMEM medium supplemented with 10% fetal bovine serum and maintained at 37 °C with 5% $CO_2$ for 0, 4, or 8 h. Cell viability was then measured by CellTiter 96® Aqueous One Solution Cell Proliferation Assay Kit (Promega, G3580).

## Statistics and reproducibility
All of the gel-based western blots are repeated for at least three times and gained similar results to the shown ones in the main manuscript or the supplementary information.

All of the results of microimaging-based immunoflurescence experiments shown in the main manuscript or the supplementary information is a representative field from at least three fields in three different times of biological replicates.

All of the mass spectrum-based proteomic experiments contains two biological replicates for each condition. All of the results of these bioligical replicates are shown in the main manuscript or the supplementary information.

The bar charts with error bar represent the mean ± SD generated from three biological replicates.

## Reporting summary
Further information on research design is available in the Nature Portfolio Reporting Summary linked to this article.

## Data availability
The proteomic data generated in this study have been deposited in the ProteomeXchange database under accession code PXD034532 (BSA invitro labeling; mitochondrial proteome; partial ERM proteome (-blue light irradiation as negative control)) and PXD037678 (partial ERM proteome (miniSOG-NES as negative control); nucleus proteome; pulse-chase ER lumen proteome in HeLa cells). The gel-based western-blot data generated in this study are provided in the Source Data file. The raw data of all the bar charts in this study are provided in the Source Data file. The PDB entries used in this work could be found in the following hyperlinks: 6GPU (miniSOG); 4F5S (BSA); 1T2W (sortase A). Source data are provided with this paper.

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

## Acknowledgements

We thank Y. Li for advice on MS sample preparation, L. Peng for assistance with image analysis, Y. Fu, Y. Zhou, and G. Wang for assistance with probe synthesis, all lab members for helpful discussions. This work was supported by the Ministry of Science and Technology (2018YFA0507600, 2017YFA0503600), the National Natural Science Foundation of China (32088101, 21727806). P.Z. is sponsored by Bayer Investigator Award. The measurement of NMR was performed at the Analytical Instrumentation Center of Peking University. We thank the Analytical Instrumentation Center in Peking University for assistance with MS sample identification and Ms. W. Zhou for help with MS results analysis.

## Author contributions

P.Z. conceived the project. F.Z. and P.Z. designed experiments. F.Z., C.Y., and X.Z. performed all experiments. F.Z. and P.Z. analyzed data and wrote the paper with inputs from other authors.

## Competing interests

The authors declare no competing interests.
