## [Peer Review File · Nature Communications]

REVIEWER COMMENTS

Reviewer #1 (Remarks to the Author):

Zheng et al. report the development of a photocatalytic labeling system consisting of genetically encoded mini-SOG that can be fused to a protein of interest for localized cellular compartment delivery and subsequent protein proximal labeling. This is achieved via use of visible light to induce localized protein oxidation that is subsequently trapped with nucleophilic tags. This method, which they call RinID, was deployed in different cellular regions for protein labeling and downstream imaging- and mass spectrometry-based analytical workflows. The authors first describe the labeling system showing it can label BSA and characterize the labeling products primarily detecting labeled histidine residues. The authors further report on optimization by testing different nucleophiles, probe concentrations and irradiation times as well as comparing to the APEX-based proximity labeling method. With their optimized conditions, they then test out mini-SOG localized to different cellular regions that include the ER, mitochondria, and nucleus. The authors are able to show strong enrichment for localized proteins within different cellular compartments that they label and compare those to other methods showing similar and/or improved protein enrichment results.

I truly appreciate that the authors showcased the strengths of their work but were also admittedly candid about possible limitations. While they show background in certain instances and openly discuss it, I don't think it will hinder this approach in being useful within mass spectrometry-based applications where these types of methods are finding the most use and impact.

Overall, this work is a very nice and natural extension to the approach the Zou lab has previously developed for mini-SOG-based oxidation and tagging of RNA in live cells (CAP-seq). This current work shows strong potential for overcoming temporal control issues and/or requirement of hydrogen peroxide associated with other enzyme-based methods. Below are some comments to address for improving the manuscript.

1) I might have missed it, but I did not see the author's previous work on CAP-seq being cited in this manuscript. This will be very important to do since it is based on essentially the same mode of activation/chemistry. As these and other methods are rapidly emerging, this will help the reader put this current approach in the appropriate context. Ideally it would be useful to do this in the introduction where other enzyme methods are being discussed. Reference 27 should also move up to the introduction as part of this discussion.

2) The authors should also cite and discuss the recently published Lux-MS method (<https://www.nature.com/articles/s41467-021-27280-x>) that uses small molecule based catalysts for achieving similar chemistry. One key strength that RINID has over LUX-MS is that the nucleophilic probe is added at the same time as the light is introduced whereas LUX-MS relies on a two step approach of oxidation followed by trapping which is a severe limitation. This should be added as a selling point for RINID.

3) Along the lines of the comment 1. The authors should consider renaming this technology to CAP-ID to match the CAP-seq approach. Given all the acronyms in this space, a similar name will be most helpful to general readers and folks within the field.

4) The background labeling in Figure 1B is quite striking. Have the authors tested other batches/lot numbers/vendors for BSA to see how widespread the background labeling is? Perhaps this also has to do with longer light exposure times, can the authors try shorter irradiation times to see if the BSA background is effected?

5) Compound numbering/labeling is a bit confusing. In some cases numbers are used and in other cases a name is used. Also, BA stands for Biotin Aniline and PA stands for propargyl amine which is very confusing because it gives the impression they are similar nucleophiles. Also, the BA and BP probes in Figure S2 panel A are swapped. Will be good to double check all of these and provide a clearer way of showing structures.

6) Also, regarding Figure S2, the western blot in panel C is quite smeary. Will be good to provide a cleaner example so the reader can better assess the differences between conditions.

7) Given the consistent use of the PA probe, the authors should provide the actual structure somewhere in the main text figures. One suggestion is to remove the structures in figure 2b since they are already in the SI Figure S2 and show the BA and BP probes since they are the most heavily used in the main manuscript.

8) Regarding the comment by the authors that:

“This difference in coverage may arise from the preferences of three methods toward different amino acid residues: whereas APEX2 and TurboID favor tyrosine and lysine, respectively, RinID targets histidine. Taken together, the above comparisons indicate that RinID offers exceptional spatial specificity and good coverage, and could complement existing methods.”

It is possible that a number of other factors are effecting the coverage including differences in the radiation time. This should be called out in addition to different residues being labeled.

9) The final main text figure falls a bit flat after the great build up and progression in the other figures. One suggestion is to move figure 5 to the SI and replace with an on/off light experiment where light is pulsed in over time to a sample and a corresponding increase in labeling is observed. This would greatly speak to the temporal control that this method seeks to achieve. I don't think an on/off experiment has ever been performed and showcased in a manuscript for light induced oxidation/trapping approaches so this would be really great to showcase.

Reviewer #2 (Remarks to the Author):

The authors developed a light-driven proximity labeling method with miniSOG. Singlet oxygen locally generated by miniSOG oxidizes the surrounding proteins, which are subsequently labeled by nucleophilic probes. This method was applied to profile several organelle proteomes and compared with other proximity labeling methods, such as APEX2 and TurboID. While proximity protein labeling using miniSOG has been performed in the past (*Bioorganic & Medicinal Chemistry Letters* 2016, 26, 3359-3363), this study is commendable in that it performed more extensive organelle proteomics. However, the idea and targeted organelles are not new, and the obtained results are also lack scientific novelty. There is also a lack of systematic and quantitative evaluation regarding labeling efficiency. In its current form, the work is incomplete and does not meet the potential significance and impact required for *Nature Communications*. It is suggested that the manuscript could be reconsidered to submit other journals after the following points have been addressed:

Major points:

1. To demonstrate novelty, applications that can only be realized with RinID should be implemented. The pulse chase labeling may be a good idea for it. But the result obtained in the present work is too preliminary and did not show the power of the method.

2. Figure 1B: The background signal in a negative control without miniSOG is very confusing. If it is due to the residual serum-derived photosensitizer impurities in the BSA, more purified BSA or other proteins should be used.

3. Figure 2B: The structure of probe1 is wrong (not the same as Figure1).

4. Figure S2: Many misconducts are found in Figure S2.

- The structure of probe1 is wrong (not the same as Figure1).

- The structure of probe 3 and probe 5 are inverse.

- It is not clear which organelle localized miniSOG is used in Figure S2B.

- Protein is not correctly transferred to the membrane in Figure S2C.

- The conditions used to evaluate each probe are different, and the analysis was not performed on the same gel, so quantitative evaluation is not possible. The authors should conduct the evaluation more systematically and show quantitative data (e.g. bar graph).

5. Figure S3: Please show the intensity plots of biotinylation levels normalized by V5 or α -tubulin signals.

6. Figure S5: Why are the positive control results (elute lane) so different for both A and B? It should be identical. Does this indicate poor experimental reproducibility?

Minor points:

1. ^{13}C -NMR for Biotin-Aniline, ^1H -NMR for Biotin-phenol, and ^{13}C -NMR for Biotin-Naphthylamine are missing.

2. MS characterization should be performed with High-resolution MS.

3. In general, LB culture medium should contain yeast extracts.

4. Please show molecular weight of marker for all gel-based analysis.

Reviewer #3 (Remarks to the Author):

Fu Zheng et al. describe a light-activated proximity-dependent intracellular labeling method to profile subcellular proteomes, termed RinID. By utilizing a genetically encoded photocatalyst, termed miniSOG, proximal proteins are modified and captured by a nucleophilic probe containing a moiety for affinity enrichment. Subsequent protein identification by mass-spectrometry revealed the capture of mitochondrial matrix proteins with high specificity. The authors assess the applicability of their technology in comparison to the well-established methodologies APEX and TurboID in various subcellular compartments including ER and nucleus.

The manuscript is in principle interesting, however, falls short of validating results and providing new biological insights. The authors show convincingly, that miniSOG's fused to a protein of interest can be used to produce singlet oxygen, which in turn can label proximal proteins - which is known for quite some time in the context of using imaging read-outs. Coupling SOG-based proximal protein tagging to an MS read-out was also recently published in NC. The screen for nucleophilic compounds identified biotin-conjugated aniline and propargylamine as highly reactive probes, which is novel and interesting from this reviewer's perspective. In summary, without providing new biological insights the manuscript would be more suitable for a more specialized, technology-focused journal.

Major points:

- Why are the authors comparing RinID with APEX instead of with the next generation APEX2 technology?

- Why are the authors using a "true negative list" (based on ribosome interactome proximity labeling) based on a non-disclosed list of their own experiments.
- H/L cutoff seems arbitrary with a non-disclosed true negative list
- The authors show enrichment for mitochondrial located proteins but do not show any validation of the found interactome or add novel biology.
- Overlap of identifications in mass spectrometry data based on duplicates does not reflect robust identification.

Minor points:

- The storyline of the paper - confusing back and forth between different probes used
- The mutated miniSOG version SOPP3, which is mentioned in the end, should have been used for the whole set of experiments.
- In the text is a reference for Fig S1e-h but the Fig. S1 only has panels a-g
- Fig. 2d V5 band shows in miniSOG but does not show up in Fig. 2c. Additionally, V5-fusion should be mentioned in the figure description.
- Fig. S3b a-tubulin loading control varies drastically plus negative control is missing for some time points.
- Fig. S8b. Replicates differ quite drastically.

We thank all three reviewers for their thorough and thoughtful comments to help us improve this manuscript. In the revised manuscript, we have provided additional experimental data on the biochemical characterization of the method, mass spec proteomic replicate data for ER membrane and the nucleus, and biological applications to profile sub-compartment protein stability in HeLa cells. We have also provided additional discussions on the advantages and disadvantages of RinID as compared to other small-molecule based proximity labeling methods. Please see our point-by-point responses below. In the revised manuscript, these changes have been marked in red.

Reviewer #1 (Remarks to the Author):

Zheng et al. report the development of a photocatalytic labeling system consisting of genetically encoded mini-SOG that can be fused to a protein of interest for localized cellular compartment delivery and subsequent protein proximal labeling. This is achieved via use of visible light to induce localized protein oxidation that is subsequently trapped with nucleophilic tags. This method, which they call RinID, was deployed in different cellular regions for protein labeling and downstream imaging- and mass spectrometry-based analytical workflows. The authors first describe the labeling system showing it can label BSA and characterize the labeling products primarily detecting labeled histidine residues. The authors further report on optimization by testing different nucleophiles, probe concentrations and irradiation times as well as comparing to the APEX-based proximity labeling method. With their optimized conditions, they then test out mini-SOG localized to different cellular regions that include the ER, mitochondria, and nucleus. The authors are able to show strong enrichment for localized proteins within different cellular compartments that they label and compare those to other methods showing similar and/or improved protein enrichment results.

I truly appreciate that the authors showcased the strengths of their work but were also admittedly candid about possible limitations. While they show background in certain instances and openly discuss it, I don't think it will hinder this approach in being useful within mass spectrometry-based applications where these types of methods are finding the most use and impact.

We thank the reviewer for these positive remarks.

Overall, this work is a very nice and natural extension to the approach the Zou lab has previously developed for mini-SOG-based oxidation and tagging of RNA in live cells (CAP-seq). This current work shows strong potential for overcoming temporal control

issues and/or requirement of hydrogen peroxide associated with other enzyme-based methods. Below are some comments to address for improving the manuscript.

1) I might have missed it, but I did not see the author's previous work on CAP-seq being cited in this manuscript. This will be very important to do since it is based on essentially the same mode of activation/chemistry. As these and other methods are rapidly emerging, this will help the reader put this current approach in the appropriate context. Ideally it would be useful to do this in the introduction where other enzyme methods are being discussed. Reference 27 should also move up to the introduction as part of this discussion.

Response: Thank you for this advice. We have re-organized the introduction part to include more discussion on existing photocatalytic proximity labeling methods, including CAP-seq and the work in reference 27.

Introduction section, paragraph 7, line 1-4: Notably, miniSOG has been used for mapping subcellular transcriptome (CAP-seq)¹⁹ and for probing protein-protein interactions²⁰, which demonstrates its high spatial specificity in labeling local biomolecules. Yet subcellular proteome-wide identification by miniSOG has not been reported.

2) The authors should also cite and discuss the recently published Lux-MS method (<https://www.nature.com/articles/s41467-021-27280-x>) that uses small molecule based catalysts for achieving similar chemistry. One key strength that RinID has over LUX-MS is that the nucleophilic probe is added at the same time as the light is introduced whereas LUX-MS relies on a two step approach of oxidation followed by trapping which is a severe limitation. This should be added as a selling point for RinID.

Response: We thank the reviewer for this suggestion. Indeed, RinID requires only one step of photocatalytic labeling, using the nucleophilic probe to intercept photo-oxidized amino acid side chains *in situ*. In the revised manuscript, we have added these discussions and comparisons with LUX-MS in the sections of Introduction and Discussion.

Introduction section, paragraph 4, line 12-14: LUX-MS applies antibody- or drug-conjugated small-molecule singlet oxygen generator and biotin-hydrazide probe to decode ligand-receptor interactions and the proteomes on cell surface¹⁷.

Discussion section, paragraph 2, line 8-10: Compared to small-molecule photocatalysts, genetically encoded RinID could be more readily targeted to

subcellular locations.

3) Along the lines of the comment 1. The authors should consider renaming this technology to CAP-ID to match the CAP-seq approach. Given all the acronyms in this space, a similar name will be most helpful to general readers and folks within the field.

Response: Thank you for this advice. While we had considered naming this method as CAP-ID, we finally decided to use RinID because we would like to emphasize the labeling mechanism (ROS-induced protein labeling) instead of the tool we used in the method (miniSOG, a chromophore) in order to distinguish our method from other chromophore-based but not ROS-based proximity labeling methods (e.g. μ -MAP (Geri, J. B., et al., *Science* **2020**, 367 (6482), 1091) and CAT-Prox (Huang, Z., et al., *JACS*. **2021**, 143 (44), 18714-18720)).

4) The background labeling in Figure 1B is quite striking. Have the authors tested other batches/lot numbers/vendors for BSA to see how widespread the background labeling is? Perhaps this also has to do with longer light exposure times, can the authors try shorter irradiation times to see if the BSA background is effected?

Response: We agree with the reviewer that the background labeling of BSA *in vitro* was indeed quite confusing. To confirm whether the labeling signal was caused by contaminant photosensitizers in the BSA we used, we repeated BSA *in vitro* labeling using another batch of BSA (New England BioLabs, B9001SVIAL). We observed similar background signal when the sample was irradiated by blue LED and labeled by Probe 1, in the absence of miniSOG. CBB staining indicated that the new source of BSA was not pure, either (Figure response 1, shown below, not in manuscript or supplemental information). We therefore turned to another protein, sortase A, which were expressed in *E. coli* and purified by ourselves as alternative choices for the *in vitro* labeling. Background labeling disappeared in the absence of miniSOG (Figure S1D). Collectively, these additional experiments confirmed that the background signal in BSA labeling *in vitro* was brought by the contaminant photosensitizers in BSA samples.

In the revised manuscript, these data are shown in Figure S1D, with the following added descriptions in the Results section, paragraph 2, line 12-15: **We repeated the above *in vitro* labeling by probe 1 with a purified protein, sortase A, and obtained similar results. No background labeling in the negative controls omitting miniSOG or light illumination was observed (Figure S1D).**

Figure response 1. In vitro labeling of BSA from another source.

Figure S1D. Western blot and CBB of miniSOG and probe 1 labeled sortase A (PDB:1T2W) samples with controls.

5) Compound numbering/labeling is a bit confusing. In some cases numbers are used and in other cases a name is used. Also, BA stands for Biotin Aniline and PA stands for propargyl amine which is very confusing because it gives the impression they are similar nucleophiles. Also, the BA and BP probes in Figure S2 panel A are swapped. Will be good to double check all of these and provide a clearer way of showing structures.

Response: Thank you for pointing out this problem. We have double-checked the structures, names and numbers of the probes throughout the text. The revised manuscript uses numbers instead of names of probes to improve readability and to remove ambiguity. For example, biotin-aniline is probe **2**, propargyl amine is probe **6**.

6) Also, regarding Figure S2, the western blot in panel C is quite smeary. Will be good to provide a cleaner example so the reader can better assess the differences between conditions.

Response: We have repeated the experiments of *in vitro* and *in vivo* comparison of different probes. In the revised manuscript, these new results are shown in Figure S2. We have also re-organized the text discussing the comparison of these probes. Please refer to the Results section, paragraph 4, line 5-19: Meanwhile, we prepared a panel of biotin-conjugated amine probes (1, 2, 4, 5), including primary alkyl amine, aniline, and hydrazide, which differ in nucleophilicity, steric hindrance and basicity (Figure 2B). For comparison, we also included biotin-conjugated phenol (probe 3), the commonly used substrate for APEX^{5, 6}. Since propargylamine has been used in CAP-seq to capture photo-oxidized guanosine, we added this probe (6) in our list of candidates as well. We tested the labeling efficiency of these probes both in the HEK293T cell lysate and live cells stably expressing cytoplasmic targeted miniSOG. For probe 6, copper(I)-catalyzed alkyne-azide cycloaddition (CuAAC) click reaction with biotin-azide was performed to install the biotin moiety prior to Western blot analysis. Whereas a similar labeling efficiency was observed for probe 6 and probes 1-4 in the cell lysate (Figure S2A), only probe 6 (propargylamine) and probe 2 (biotin-aniline) yielded strong labeling signal in living cells (Figure S2B). We speculated that probes 1, 3, and 4 may have limited permeability through the cell membrane, while probe 5 lacks the ability to capture photo-oxidized protein intermediates.

Figure S2. Comparison of different probes for RinID activity in live cells and in cell lysates. **A)** Western blot and CBB of miniSOG labeled proteome samples with different nucleophilic probes in cell lysate; Probe concentration: 10 mM probe 1 and 6, 0.5 mM for other four probes. **B)** Western blot of miniSOG labeled proteome samples with different nucleophilic probes in miniSOG-NES HEK293T stable cell line. Probe concentration: 10 mM probe 1 and 6, 0.5 mM for other four probes. Bar graphs indicates the signal intensity of Streptavidin-HRP blot of each lane normalized by CBB. The concentration of SDS-PAGE is 4-20%.

7) Given the consistent use of the PA probe, the authors should provide the actual structure somewhere in the main text figures. One suggestion is to remove the structures in figure 2b since they are already in the SI Figure S2 and show the BA and BP probes since they are the most heavily used in the main manuscript.

Response: In the revised manuscript, we have added the chemical structure of propargyl amine (probe 6) in Figure 2B. The structure of PA is also shown in the experimental scheme in Figure 1. We have decided to include the structures of all probes in the main text, for helping explaining the results of comparisons.

Figure 2B. Chemical structures of probes used in this study.

8) Regarding the comment by the authors that:

“This difference in coverage may arise from the preferences of three methods toward different amino acid residues: whereas APEX2 and TurboID favor tyrosine and lysine, respectively, RinID targets histidine. Taken together, the above comparisons indicate that RinID offers exceptional spatial specificity and good coverage, and could complement existing methods.”

It is possible that a number of other factors are effecting the coverage including differences in the radiation time. This should be called out in addition to different residues being labeled.

Response: Thank you for this good advice. In the revised manuscript, we have added additional discussion about the difference in coverage of three proximity labeling methods including labeling mechanisms, labeling time, and the strategy chosen for quantitative proteomics identification.

Results section, paragraph 11, line 17-20: In addition, variations in the labeling protocols (photocatalytic vs. enzymatic) and the quantitative proteomics analysis methods could also contribute to the difference in coverage.

9) The final main text figure falls a bit flat after the great build up and progression in the other figures. One suggestion is to move figure 5 to the SI and replace with an on/off light experiment where light is pulsed in over time to a sample and a

corresponding increase in labeling is observed. This would greatly speak to the temporal control that this method seeks to achieve. I don't think an on/off experiment has ever been performed and showcased in a manuscript for light induced oxidation/trapping approaches so this would be really great to showcase.

Response: Thank you for this advice. To highlight the advantage of temporal control of RinID, we have performed a pulse-chase labeling experiment in HeLa cells to quantify the degradation rate of a specific cohort of proteins with subcellular resolution. Such analysis could not be achieved with conventional methods, including APEX-based proximity labeling. We applied quantitative proteomics by TMT-based MS identification to compare the degradation speed of different proteins in ER lumen (Figure 5, Figure S8). Our data revealed high heterogeneity in the clearance rates for proteins in the ER lumen, with secreted proteins cleared faster than ER resident proteins. Our results could shed light on how cells regulate protein stability and degradation. In the revised manuscript, we have added the following text describing this experiment:

Results section, paragraphs 16-18: We thus design a pulse-chase labeling scheme in HeLa cells stably expressing ER lumen targeted SOPP3 to monitor the clearance rate of secretory pathway proteins by TMT labeling-based quantitative proteomics. Cells were labeled with 5 mM probe **6** and blue LED illumination at 30 mW·cm⁻² for 5 min, which was followed by chasing in normal cell culture medium for up to 24 hours. Cells were sampled at 8 hours intervals, clicked with biotin-azide, and enriched with streptavidin-coated agarose beads. Western blot and silver staining indicated that the labeling signal decreased dramatically after 16 hours chasing (Figure 5B, Figure S8D). Enriched proteins were digested by trypsin and labeled with TMT reagents for isobaric quantitative LC-MS/MS analysis (Figure 5A). To account for the slight variations in sample loading, we normalized the TMT reporter ion intensity of each protein with respect to the signal of an endogenously biotinylated protein, pyruvate carboxylase.

To determine the ER lumen proteome, we calculated the ratios of the averaged reporter ion intensity of two replicates of 0 hour sample (126 and 127N) over the negative controls omitting either the probe (130C) or the blue light illumination (131). For ROC analysis, we used the same 'true positive' and 'false positive' lists as those in the previous ERM analysis (Table S5-2, S5-3), which revealed cut-off log₂ ratios as 2.44 and 1.44 (Figure S8E) for +/- probe and +/- light, respectively. We further filtered out proteins with averaged intensities less than 10000 in the 0 hour samples (126 and 127N), yielding a final list of 100 proteins. Notably, 97 out of 100 have secretory pathway annotations in Gene Ontology, indicating high spatial specificity of RinID in the ER lumen (Figure S8F, Table S5-4).

Protein abundance at each time point in the chase period was normalized with respect to the initial state of 0 hour. We calculated the ratios of the averaged reporter ion intensity of each protein at 8 hours, 16 hours, and 24 hours over those at 0 hour (Figure 5C, Table S5-4). As expected, an overall downward trend was observed for protein abundance as a function of chase time (A-t curve), with an averaged decrease of $7.0\pm 15.6\%$, $41.5\pm 10.5\%$ and $45.7\pm 10.6\%$ at 8 hours, 16 hours and 24 hours, respectively. Within this decreasing background, there were considerable variations in the clearance rate among individual proteins. To quantitatively measure protein clearance, we defined Retention Index (RI) for each protein as the area under its A-t curve (Table S5-4). Figure 5D ranks proteins according to their RI. Notably, a few proteins exhibit substantially lower RI, indicating faster clearance. Gene Ontology annotations reveal terms related to secretion for these proteins (Figure 5D). Indeed, among the 12 secreted proteins in our dataset, 6 are rapidly cleared from cells with RI lower than 0.70. The fastest two proteins, TGM3 and LIPL, are already removed by 73% and 51% at 8 hours. In contrast, ER resident proteins typically have higher RI, with an average of 0.79 ± 0.06 for the 9 proteins in our dataset. The observed variations in protein clearance dynamics attests to the complexity of the mechanisms by which cells regulate proteomic homeostasis. Through the above proof-of-concept experiment, we demonstrate the feasibility of monitoring subcellular proteomic dynamics with RinID, which is the first proximity labeling method that is compatible with pulse-chase labeling scheme.

Figure 5. Pulse-chase labeling of secretory pathway proteome with RinID. **A)** scheme of pulse-chase RinID labeling and TMT-based quantitative proteomics. **B)** Western blot of the labeling signal at different chasing time point. SDS-PAGE concentration: 4-20%. **C)** Decrease ratio of RinID identified ER lumen proteins at different chasing time point compared to 0 hour chasing. Blue line indicates the average of the identified proteins. **D)** Retention rate of RinID identified ER lumen proteins. Different colors indicate different kind of proteins. Red: secreted proteins; Green: ER resident proteins; Magenta: ER resident with secreted subcellular location annotation.

Figure S8. Pulse-chase labeling of secretory pathway proteome with SOPP3-based RinID.
A) Western blot of ER lumen proteome of HeLa cells labeled with miniSOG or SOPP3 at different probe 6 concentrations, with blue light illumination at 30 mW·cm⁻² for 5 min. **B)** Cell viability

assay of HeLa cells at 0, 4, and 8 hours after SOPP3 labeling in the ER lumen with 5 min blue light irradiation and 5 mM probe 6. **C)** Confocal fluorescence images comparing miniSOG and SOPP3 labeling in the ER lumen of HeLa cells. miniSOG and SOPP3 are targeted to the ER lumen via N-terminal fusion of Igk secretory sequence and C-terminal fusion of KDEL motif (ss-miniSOG/SOPP3-KDEL). Calnexin is a marker for ER. Labeling condition: 5 mM probe 6, 30 mW·cm⁻², 5 min blue LED irradiation. Scale bar: 20 μm. **D)** Silver staining of ER lumen proteome samples of HeLa cells pulse-chase labeled by ss-SOPP3-KDEL for MS identification, with negative control omitting probe 6 or blue light. The concentration of SDS-PAGE is 4-20%. **E)** Receiver operator curve (ROC) analysis of ER lumen proteome identified by pulse-chase RinID. ROC curves of the proteins ranked by $\log_2((126+127N)/(2*130C))$ (chase 0 hour/- probe 6) and by $\log_2((126+127N)/(2*131))$ (chase 0 hour/- blue light) are used to determine cut-off ratio. The FPR and TPR of the cut-off points are shown as red dots. The cut-off ratio and the TPR-FPR at the cut-off point are shown as blue dots. **F)** Spatial specificity analysis of ER lumen proteome identified by SOPP3 mediated pulse-chase RinID.

Reviewer #2 (Remarks to the Author):

The authors developed a light-driven proximity labeling method with miniSOG. Singlet oxygen locally generated by miniSOG oxidizes the surrounding proteins, which are subsequently labeled by nucleophilic probes. This method was applied to profile several organelle proteomes and compared with other proximity labeling methods, such as APEX2 and TurboID. While proximity protein labeling using miniSOG has been performed in the past (Bioorganic & Medicinal Chemistry Letters 2016, 26, 3359-3363), this study is commendable in that it performed more extensive organelle proteomics. However, the idea and targeted organelles are not new, and the obtained results are also lack scientific novelty. There is also a lack of systematic and quantitative evaluation regarding labeling efficiency. In its current form, the work is incomplete and does not meet the potential significance and impact required for Nature Communications. It is suggested that the manuscript could be reconsidered to submit other journals after the following points have been addressed:

Major points:

1. To demonstrate novelty, applications that can only be realized with RinID should be implemented. The pulse chase labeling may be a good idea for it. But the result obtained in the present work is too preliminary and did not show the power of the method.

Response: We thank the reviewer for this suggestion. In the revised manuscript, we

have provided additional experimental data on pulse-chase labeling of ER lumen proteome with RinID (Figure 5, Figure S8). This experiment aims to quantify the degradation rate of a specific cohort of proteins with subcellular resolution, and such analysis could not be achieved with conventional methods, including APEX-based proximity labeling. Our data revealed high heterogeneity in the clearance rates for proteins in the ER lumen, with secreted proteins cleared faster than ER resident proteins. Our results could shed light on how cells regulate protein stability and degradation. In the revised manuscript, we have added the following text describing this experiment:

Results section, paragraphs 16-18: We thus design a pulse-chase labeling scheme in HeLa cells stably expressing ER lumen targeted SOPP3 to monitor the clearance rate of secretory pathway proteins by TMT labeling-based quantitative proteomics. Cells were labeled with 5 mM probe **6** and blue LED illumination at 30 mW·cm⁻² for 5 min, which was followed by chasing in normal cell culture medium for up to 24 hours. Cells were sampled at 8 hours intervals, clicked with biotin-azide, and enriched with streptavidin-coated agarose beads. Western blot and silver staining indicated that the labeling signal decreased dramatically after 16 hours chasing (Figure 5B, Figure S8D). Enriched proteins were digested by trypsin and labeled with TMT reagents for isobaric quantitative LC-MS/MS analysis (Figure 5A). To account for the slight variations in sample loading, we normalized the TMT reporter ion intensity of each protein with respect to the signal of an endogenously biotinylated protein, pyruvate carboxylase.

To determine the ER lumen proteome, we calculated the ratios of the averaged reporter ion intensity of two replicates of 0 hour sample (126 and 127N) over the negative controls omitting either the probe (130C) or the blue light illumination (131). For ROC analysis, we used the same 'true positive' and 'false positive' lists as those in the previous ERM analysis (Table S5-2, S5-3), which revealed cut-off log₂ ratios as 2.44 and 1.44 (Figure S8E) for +/- probe and +/- light, respectively. We further filtered out proteins with averaged intensities less than 10000 in the 0 hour samples (126 and 127N), yielding a final list of 100 proteins. Notably, 97 out of 100 have secretory pathway annotations in Gene Ontology, indicating high spatial specificity of RinID in the ER lumen (Figure S8F, Table S5-4).

Protein abundance at each time point in the chase period was normalized with respect to the initial state of 0 hour. We calculated the ratios of the averaged reporter ion intensity of each protein at 8 hours, 16 hours, and 24 hours over those at 0 hour (Figure 5C, Table S5-4). As expected, an overall downward trend was observed for protein abundance as a function of chase time (A-t curve), with an averaged decrease of 7.0±15.6%, 41.5±10.5% and 45.7±10.6% at 8 hours, 16 hours and 24 hours, respectively. Within this decreasing background, there were considerable variations in

the clearance rate among individual proteins. To quantitatively measure protein clearance, we defined Retention Index (RI) for each protein as the area under its A-t curve (Table S5-4). Figure 5D ranks proteins according to their RI. Notably, a few proteins exhibit substantially lower RI, indicating faster clearance. Gene Ontology annotations reveal terms related to secretion for these proteins (Figure 5D). Indeed, among the 12 secreted proteins in our dataset, 6 are rapidly cleared from cells with RI lower than 0.70. The fastest two proteins, TGM3 and LIPL, are already removed by 73% and 51% at 8 hours. In contrast, ER resident proteins typically have higher RI, with an average of 0.79 ± 0.06 for the 9 proteins in our dataset. The observed variations in protein clearance dynamics attests to the complexity of the mechanisms by which cells regulate proteomic homeostasis. Through the above proof-of-concept experiment, we demonstrate the feasibility of monitoring subcellular proteomic dynamics with RinID, which is the first proximity labeling method that is compatible with pulse-chase labeling scheme.

Figure 5. Pulse-chase labeling of secretory pathway proteome with RinID. **A)** scheme of pulse-chase RinID labeling and TMT-based quantitative proteomics. **B)** Western blot of the labeling

signal at different chasing time point. SDS-PAGE concentration: 4-20%. **C)** Decrease ratio of RinID identified ER lumen proteins at different chasing time point compared to 0 hour chasing. Blue line indicates the average of the identified proteins. **D)** Retention rate of RinID identified ER lumen proteins. Different colors indicate different kind of proteins. Red: secreted proteins; Green: ER resident proteins; Magenta: ER resident with secreted subcellular location annotation.

Figure S8. Pulse-chase labeling of secretory pathway proteome with SOPP3-based RinID.
A) Western blot of ER lumen proteome of HeLa cells labeled with miniSOG or SOPP3 at different probe 6 concentrations, with blue light illumination at 30 mW·cm⁻² for 5 min. **B)** Cell viability

assay of HeLa cells at 0, 4, and 8 hours after SOPP3 labeling in the ER lumen with 5 min blue light irradiation and 5 mM probe 6. **C)** Confocal fluorescence images comparing miniSOG and SOPP3 labeling in the ER lumen of HeLa cells. miniSOG and SOPP3 are targeted to the ER lumen via N-terminal fusion of Igk secretory sequence and C-terminal fusion of KDEL motif (ss-miniSOG/SOPP3-KDEL). Calnexin is a marker for ER. Labeling condition: 5 mM probe 6, 30 mW·cm⁻², 5 min blue LED irradiation. Scale bar: 20 μm. **D)** Silver staining of ER lumen proteome samples of HeLa cells pulse-chase labeled by ss-SOPP3-KDEL for MS identification, with negative control omitting probe 6 or blue light. The concentration of SDS-PAGE is 4-20%. **E)** Receiver operator curve (ROC) analysis of ER lumen proteome identified by pulse-chase RinID. ROC curves of the proteins ranked by $\log_2((126+127N)/(2*130C))$ (chase 0 hour/- probe 6) and by $\log_2((126+127N)/(2*131))$ (chase 0 hour/- blue light) are used to determine cut-off ratio. The FPR and TPR of the cut-off points are shown as red dots. The cut-off ratio and the TPR-FPR at the cut-off point are shown as blue dots. **F)** Spatial specificity analysis of ER lumen proteome identified by SOPP3 mediated pulse-chase RinID.

2. Figure 1B: The background signal in a negative control without miniSOG is very confusing. If it is due to the residual serum-derived photosensitizer impurities in the BSA, more purified BSA or other proteins should be used.

Response: We thank the reviewer for pointing out this issue. The background signal of BSA *in vitro* labeling indeed confused us. As mentioned in our response to referee #1, we have repeated the BSA *in vitro* labeling with another batch of BSA sample of BSA (New England BioLabs, B9001SVIAL). We observed similar background signal when the sample was irradiated by blue LED and labeled by Probe 1, in the absence of miniSOG. Coomassie staining indicated the presence of contaminant proteins in this new source of BSA (Figure response 1, shown below, not in manuscript or supplemental information), which may contain photosensitizers. We therefore chose another protein, sortase A, which, we know for sure, is not a photosensitizer. We expressed the protein in *E. coli* and purified it before performing *in vitro* labeling. Background labeling disappeared in the absence of miniSOG (Figure S1D). Collectively, these additional experiments confirmed that the background signal in BSA labeling *in vitro* was brought by the contaminant photosensitizers in BSA samples. In the revised manuscript, these data are shown in Figure S1D, with the following added descriptions in the Results section, paragraph 2, line 12-15: We repeated the above *in vitro* labeling by probe 1 with a purified protein, sortase A, and obtained similar results. No background labeling in the negative controls omitting miniSOG or light illumination was observed (Figure S1D).

Figure response 1. In vitro labeling of BSA from another source.

Figure S1D. Western blot and CBB of miniSOG and probe 1 labeled sortase A (PDB:1T2W) samples with controls.

3. Figure 2B: The structure of probe1 is wrong (not the same as Figure1).

Response: Thank you for pointing out this problem. We have double-checked all of the structures, names, and numbers of the probes throughout the revised manuscript to enhance readability and reduce ambiguity.

Figure 2B. Chemical structures of probes used in this study.

4. Figure S2: Many misconducts are found in Figure S2.

- The structure of probe 1 is wrong (not the same as Figure 1).
- The structure of probe 3 and probe 5 are inverse.
- It is not clear which organelle localized miniSOG is used in Figure S2B.
- Protein is not correctly transferred to the membrane in Figure S2C.
- The conditions used to evaluate each probe are different, and the analysis was not performed on the same gel, so quantitative evaluation is not possible. The authors should conduct the evaluation more systematically and show quantitative data (e.g. bar graph).

Response: Thank you for pointing out these issues. In addition to the improvement of our manuscript mentioned in point #3, we have repeated the experiments of *in vitro* and *in vivo* comparisons of different probes and added quantitative analysis of the blot images. These new results are shown in the revised Figure S2. In the revised manuscript text, we have also rewritten this part of discussion in the Results section, paragraph 4, line 5-19: Meanwhile, we prepared a panel of biotin-conjugated amine probes (1, 2, 4, 5), including primary alkyl amine, aniline, and hydrazide, which differ in nucleophilicity, steric hindrance and basicity (Figure 2B). For comparison, we also included biotin-conjugated phenol (probe 3), the commonly used substrate for APEX⁵.⁶ Since propargylamine has been used in CAP-seq to capture photo-oxidized guanosine, we added this probe (6) in our list of candidates as well. We tested the labeling efficiency of these probes both in the HEK293T cell lysate and live cells stably expressing cytoplasmic targeted miniSOG. For probe 6, copper(I)-catalyzed alkyne-azide cycloaddition (CuAAC) click reaction with biotin-azide was performed to install

the biotin moiety prior to Western blot analysis. Whereas a similar labeling efficiency was observed for probe 6 and probes 1-4 in the cell lysate (Figure S2A), only probe 6 (propargylamine) and probe 2 (biotin-aniline) yielded strong labeling signal in living cells (Figure S2B). We speculated that probes 1, 3, and 4 may have limited permeability through the cell membrane, while probe 5 lacks the ability to capture photo-oxidized protein intermediates.¹

Figure S2. Comparison of different probes for RinID activity in live cells and in cell lysates. **A)** Western blot and CBB of miniSOG labeled proteome samples with different nucleophilic probes in cell lysate; Probe concentration: 10 mM probe 1 and 6, 0.5 mM for other four probes. **B)** Western blot of miniSOG-NES HEK293T stable cell line. Probe concentration: 10 mM probe 1 and 6, 0.5 mM for other four probes. Bar graphs indicates the signal intensity of Streptavidin-HRP blot of each lane normalized by CBB. The concentration of SDS-PAGE is 4-20%.

5. Figure S3: Please show the intensity plots of biotinylation levels normalized by V5 or a-tubulin signals.

Response: Thank you for this advice. In the revised manuscript, the intensities of SA-HRP blots of Figures S3A and S3B have been normalized with respect to the CBB signal (as a measure of cell lysate loading) and V5 signal (as a measure of miniSOG expression), respectively. The intensity plots are shown in revised Figure S3.

Figure S3. Optimization of probe 6 concentration and illumination time. **A-B)** Western blot analysis of proteins labeled with mitochondrial matrix-targeted miniSOG (mito-V5-miniSOG) at different probe 6 concentrations (A) and illumination time (B). Bar graphs indicates the signal intensity of Streptavidin-HRP blot of each lane normalized by V5 (only miniSOG+ lanes are analyzed in (B)). The concentration of SDS-PAGE is 4-20%.

6. Figure S5: Why are the positive control results (elute lane) so different for both A and B? It should be identical. Does this indicate poor experimental reproducibility?

Response: Thank you for pointing out this issue. We apologize for not explaining this earlier in the text. The polyacrylamide gel used for SDS-PAGE in Figures S5A and S5B contain different percentages of acrylamide/bisacrylamide, which is why the protein bands in gel images appear different. The experiment of figure S5A used 12% SDS-PAGE, while the experiment of figure S5B used 4-20% concentration gradient SDS-PAGE. Nevertheless, these differences do not change the conclusions drawn from these experiments. In fact, our proteomic data have shown good reproducibility

between replicates of labeling (Figure 3, table S2). In the revised manuscript, we have explicitly stated the differences in the polyacrylamide concentration used in these experiments in the explanatory text of each figure containing SDS-PAGE related results, to avoid confusing the readers.

Minor points:

1. ^{13}C -NMR for Biotin-Aniline, ^1H -NMR for Biotin-phenol, and ^{13}C -NMR for Biotin-Naphthylamine are missing.

Response: In the revised manuscript, we have shown the NMR data for Biotin-Aniline and Biotin-phenol. Unfortunately, we do not have sufficient Biotin-Naphthylamine for producing a high quality ^{13}C -NMR spectra. Given the difficulty of its synthesis and its poor performance in labeling, we have decided not to use this probe and have removed it from comparisons in the revised manuscript.

Supporting information, page 6-7: ^1H -NMR for Biotin-Aniline (400 MHz, d^6 -DMSO): 7.78 (1H, t), 6.84 (2H, d), 6.49 (2H, d), 6.43 (1H, s), 6.36 (1H, s), 4.85 (2H, s), 4.31 (1H, m), 4.12 (1H, m), 3.16 (2H, dd), 3.09 (2H, dd), 2.84 (1H, dd), 2.56 (3H, m), 2.03 (2H, t), 1.38-1.66 (4H, m), 1.28 (2H, m). ^{13}C -NMR for Biotin-Aniline (100 MHz, d^6 -DMSO): 25.79, 28.51, 28.65, 35.00, 35.69, 40.32, 41.07, 55.89, 59.66, 61.50, 114.42, 126.82, 129.40, 147.18, 163.18, 172.27. ^1H -NMR for Biotin-phenol (500 MHz, d^6 -DMSO): 7.85 (1H, t), 6.96 (2H, d), 6.67 (2H, d), 6.56 (1H, s), 6.46 (1H, s), 4.30 (1H, m), 4.12 (1H, m), 3.18 (2H, dd), 3.05 (1H, dd), 2.79 (1H, dd), 2.57 (3H, m), 2.04 (2H, t), 1.42-1.61 (4H, m), 1.27 (2H, m). ^{13}C -NMR for Biotin-phenol (125 MHz, d^6 -DMSO): 25.50, 28.18, 28.36, 34.56, 35.40, 40.06, 55.63, 59.46, 61.28, 115.28, 129.62, 140.70, 155.92, 163.16, 172.29, 192.48.

2. MS characterization should be performed with High-resolution MS.

Response: For the MS characterization of Biotin-Aniline and Biotin-phenol, we used Fourier Transform High Resolution Mass Spectrometry (FTMS) in the revised manuscript to gain higher resolution.

Supporting information, page 7: The following MS characterization of purified probes was acquired on Fourier Transform High Resolution Mass Spectrometry (FTMS) :

Biotin-Aniline: calculated for $\text{C}_{18}\text{H}_{27}\text{N}_4\text{O}_2\text{S}$: $[\text{M}+\text{H}]^+$: 363.18547; found: 363.18492

Biotin-Phenol: calculated for $\text{C}_{18}\text{H}_{26}\text{N}_3\text{O}_3\text{S}$: $[\text{M}+\text{H}]^+$: 364.16949; found: 364.16894

For the LC-MS/MS analysis of proteome samples, the MS resolution is indeed an important parameter for proteomic analysis. The resolution of Orbitrap Fusion LUMOS Tribid Mass Spectrometer used in our experiment is 60000 (MS) and 15000 (MS/MS). We believe this resolution could match our demand for proof of concept of RinID in our current work. We may consider to use MS with higher resolution in further experiments which needs higher coverage. In the revised Supplementary methods, we have added information on the MS resolution.

Supporting information, page 15: Survey scans of peptide precursors were collected in the Orbitrap from 350-1600 Th with an AGC target of 400,000, a maximum injection time of 50 ms, RF lens at 30%, and a resolution of 60,000 at 200 m/z. Monoisotopic precursor selection was enabled for peptide isotopic distributions, precursors of z = 2-7 were selected for data-dependent MS/MS scans (with a resolution of 15000) for 3 seconds of cycle time, and dynamic exclusion was set to 15 seconds with a ± 10 ppm window set around the precursor mono-isotope.

3. In general, LB culture medium should contain yeast extracts.

Response: Thank you for pointing out this mistake. We have corrected the formula of LB culture medium in the revised supporting information.

Supporting information, page 7: LB culture medium: 10 g NaCl, 10 g tryptone, 5 g yeast extract in 1 L ddH₂O

4. Please show molecular weight of marker for all gel-based analysis.

Response: Thank you for this advice. In the revised manuscript, we have indicated the molecular weight of protein markers in the following gel images: figure 1B, figure 5B, figure S1C-D, figure S2 B-C, figure S5A-B, figure S6B, figure S7C, figure S8A, and figure S8D.

Reviewer #3 (Remarks to the Author):

Fu Zheng et al. describe a light-activated proximity-dependent intracellular labeling method to profile subcellular proteomes, termed RinID. By utilizing a genetically encoded photocatalyst, termed miniSOG, proximal proteins are modified and captured by a nucleophilic probe containing a moiety for affinity enrichment. Subsequent protein

identification by mass-spectrometry revealed the capture of mitochondrial matrix proteins with high specificity. The authors assess the applicability of their technology in comparison to the well-established methodologies APEX and TurboID in various subcellular compartments including ER and nucleus.

The manuscript is in principle interesting, however, falls short of validating results and providing new biological insights. The authors show convincingly, that miniSOG's fused to a protein of interest can be used to produce singlet oxygen, which in turn can label proximal proteins - which is known for quite some time in the context of using imaging read-outs. Coupling SOG-based proximal protein tagging to an MS read-out was also recently published in NC. The screen for nucleophilic compounds identified biotin-conjugated aniline and propargylamine as highly reactive probes, which is novel and interesting from this reviewer's perspective. In summary, without providing new biological insights the manuscript would be more suitable for a more specialized, technology-focused journal.

Major points:

- Why are the authors comparing RinID with APEX instead of with the next generation APEX2 technology?

Response: The MS data of APEX (mitochondria), APEX2 (ERM), and TurboID (mitochondria, ERM, nucleus) used in the manuscript are all previously published in other works. Though APEX2 is the next generation version of APEX, there is no published data for APEX2-based mitochondria matrix proteomics data to the best of our knowledge. This may be because that the APEX-based mitochondria matrix proteomics data is good enough (which has been used as a reliable source in mitocarta 3.0 database). Thus, we have decided to compare our RinID dataset to published APEX-based mitochondria matrix proteomics dataset. For other subcellular compartments, we have compared RinID dataset against APEX2 datasets, which are available in the literature.

- Why are the authors using a "true negative list" (based on ribosome interactome proximity labeling) based on a non-disclosed list of their own experiments.

Response: We thank the reviewer for pointing out this issue. The ribosome interactome data came from a high confident MS experiment based on BirA-mediated ribosome proximity biotinylation by our lab. None of the proteins in the list appears in

mitocarta 3.0 with 'matrix' or 'MIM' annotation. We now realize that this dataset may confuse the readers and therefore decide to change the 'true positive list' and 'true negative list' to the same ones used in the TurboID paper (PubMed ID: 30125270, table S2-3, S2-4). The results have been updated in the revised manuscript and supplemental tables. The specificity and coverage changed only a little bit, from 394 (97%) to 477 (94%).

- H/L cutoff seems arbitrary with a non-disclosed true negative list

Response: In the experiment to identify the mitochondria proteome, the H/L cutoff is prudently selected due to the result of ROC analysis to gain both high spatial specificity and proteome coverage. In the experiment about ERM and nucleus proteome, we have taken additional replicates with different negative controls to improve the reliability of our proteome data. To set an H/L cutoff ratio reasonably, we also take ROC analysis by the 'true positive list' and 'true negative list' used in the research of TurboID (Table S3-3, 3-4, 4-3, 4-4). Under the cutoff we have set, the proteomic data indicate higher specificity of RinID (93% and 92%) than TurboID (72% and 79%), though the coverage of our method is much lower. However, the relatively lower coverage may be caused by the process of MS identification, but not the proximity labeling itself. In our opinion, the current H/L cutoff ratio from ROC analysis does give a high enough spatial specificity to our proteome dataset, which is the most important criterion to evaluate a proximity method when the coverage is acceptable.

- The authors show enrichment for mitochondrial located proteins but do not show any validation of the found interactome or add novel biology.

Response: Although proteome identification in mitochondria is just a proof of concept in our work, we have shown the topology of proteins identified on the electron-transport chain complexes on IMM to demonstrate the high spatial specificity of RinID. Indeed, there is not novel biology from our MS identification in mitochondria, ERM, and nucleus. To further show the power of RinID, we expand the pulse-chase labeling in HeLa cells to quantitative proteomics by TMT-based MS identification to compare the degradation speed of different proteins in ER lumen (Figure 5, Figure S8). Our data revealed high heterogeneity in the clearance rates for proteins in the ER lumen, with secreted proteins cleared faster than ER resident proteins. We believe that this could not be realized by existing proximity labeling methods. Our results could shed light on how cells regulate protein stability and degradation. In the revised manuscript, we have added the following text describing this experiment in the Results section, paragraphs 16-18: **We thus design a pulse-chase labeling scheme in Hela cells stably expressing**

ER lumen targeted SOPP3 to monitor the clearance rate of secretory pathway proteins by TMT labeling-based quantitative proteomics. Cells were labeled with 5 mM probe 6 and blue LED illumination at $30 \text{ mW} \cdot \text{cm}^{-2}$ for 5 min, which was followed by chasing in normal cell culture medium for up to 24 hours. Cells were sampled at 8 hours intervals, clicked with biotin-azide, and enriched with streptavidin-coated agarose beads. Western blot and silver staining indicated that the labeling signal decreased dramatically after 16 hours chasing (Figure 5B, Figure S8D). Enriched proteins were digested by trypsin and labeled with TMT reagents for isobaric quantitative LC-MS/MS analysis (Figure 5A). To account for the slight variations in sample loading, we normalized the TMT reporter ion intensity of each protein with respect to the signal of an endogenously biotinylated protein, pyruvate carboxylase.

To determine the ER lumen proteome, we calculated the ratios of the averaged reporter ion intensity of two replicates of 0 hour sample (126 and 127N) over the negative controls omitting either the probe (130C) or the blue light illumination (131). For ROC analysis, we used the same 'true positive' and 'false positive' lists as those in the previous ERM analysis (Table S5-2, S5-3), which revealed cut-off \log_2 ratios as 2.44 and 1.44 (Figure S8E) for +/- probe and +/- light, respectively. We further filtered out proteins with averaged intensities less than 10000 in the 0 hour samples (126 and 127N), yielding a final list of 100 proteins. Notably, 97 out of 100 have secretory pathway annotations in Gene Ontology, indicating high spatial specificity of RinID in the ER lumen (Figure S8F, Table S5-4).

Protein abundance at each time point in the chase period was normalized with respect to the initial state of 0 hour. We calculated the ratios of the averaged reporter ion intensity of each protein at 8 hours, 16 hours, and 24 hours over those at 0 hour (Figure 5C, Table S5-4). As expected, an overall downward trend was observed for protein abundance as a function of chase time (A-t curve), with an averaged decrease of $7.0 \pm 15.6\%$, $41.5 \pm 10.5\%$ and $45.7 \pm 10.6\%$ at 8 hours, 16 hours and 24 hours, respectively. Within this decreasing background, there were considerable variations in the clearance rate among individual proteins. To quantitatively measure protein clearance, we defined Retention Index (RI) for each protein as the area under its A-t curve (Table S5-4). Figure 5D ranks proteins according to their RI. Notably, a few proteins exhibit substantially lower RI, indicating faster clearance. Gene Ontology annotations reveal terms related to secretion for these proteins (Figure 5D). Indeed, among the 12 secreted proteins in our dataset, 6 are rapidly cleared from cells with RI lower than 0.70. The fastest two proteins, TGM3 and LIPL, are already removed by 73% and 51% at 8 hours. In contrast, ER resident proteins typically have higher RI, with an average of 0.79 ± 0.06 for the 9 proteins in our dataset. The observed variations in protein clearance dynamics attests to the complexity of the mechanisms by which cells regulate proteomic homeostasis. Through the above proof-of-concept experiment,

we demonstrate the feasibility of monitoring subcellular proteomic dynamics with RinID, which is the first proximity labeling method that is compatible with pulse-chase labeling scheme.

Figure 5. Pulse-chase labeling of secretory pathway proteome with RinID. **A)** scheme of pulse-chase RinID labeling and TMT-based quantitative proteomics. **B)** Western blot of the labeling signal at different chasing time point. SDS-PAGE concentration: 4-20%. **C)** Decrease ratio of RinID identified ER lumen proteins at different chasing time point compared to 0 hour chasing. Blue line indicates the average of the identified proteins. **D)** Retention rate of RinID identified ER lumen proteins. Different colors indicate different kind of proteins. Red: secreted proteins; Green: ER resident proteins; Magenta: ER resident with secreted subcellular location annotation.

Figure S8. Pulse-chase labeling of secretory pathway proteome with SOPP3-based RinID.
A) Western blot of ER lumen proteome of HeLa cells labeled with miniSOG or SOPP3 at different probe 6 concentrations, with blue light illumination at 30 mW·cm⁻² for 5 min. **B)** Cell viability

assay of HeLa cells at 0, 4, and 8 hours after SOPP3 labeling in the ER lumen with 5 min blue light irradiation and 5 mM probe 6. **C)** Confocal fluorescence images comparing miniSOG and SOPP3 labeling in the ER lumen of HeLa cells. miniSOG and SOPP3 are targeted to the ER lumen via N-terminal fusion of Igk secretory sequence and C-terminal fusion of KDEL motif (ss-miniSOG/SOPP3-KDEL). Calnexin is a marker for ER. Labeling condition: 5 mM probe 6, 30 mW·cm⁻², 5 min blue LED irradiation. Scale bar: 20 μm. **D)** Silver staining of ER lumen proteome samples of HeLa cells pulse-chase labeled by ss-SOPP3-KDEL for MS identification, with negative control omitting probe 6 or blue light. The concentration of SDS-PAGE is 4-20%. **E)** Receiver operator curve (ROC) analysis of ER lumen proteome identified by pulse-chase RinID. ROC curves of the proteins ranked by $\log_2((126+127N)/(2*130C))$ (chase 0 hour/- probe 6) and by $\log_2((126+127N)/(2*131))$ (chase 0 hour/- blue light) are used to determine cut-off ratio. The FPR and TPR of the cut-off points are shown as red dots. The cut-off ratio and the TPR-FPR at the cut-off point are shown as blue dots. **F)** Spatial specificity analysis of ER lumen proteome identified by SOPP3 mediated pulse-chase RinID.

- Overlap of identifications in mass spectrometry data based on duplicates does not reflect robust identification.

Response: In the mitochondria proteome identification MS experiment, we have set four biological replicates (two with omitting blue LED irradiation as negative control, while two with omitting miniSOG). H/L cut off was set by ROC analysis (the true positive list and true negative list has been updated to the lists used in the research of TurboID) and the proteins passed the cut off in all of the four replicates were combined as our mitochondria proteome (477 proteins, 450 of them appear in mitocarta 3.0 database, 94% specificity) (Figure 3C). We believe the high coverage and spatial specificity of this experiment do reflect robust identification.

In the previous version of the manuscript, proteome identification at ERM and nucleus lack some groups of replicates. To enhance the reliability of our results, we took additional MS identifications. For the ERM proteome, we took two additional replicates with cytoplasm located miniSOG (miniSOG-NES) as negative control. H/L cut off was set by ROC analysis (list from TurboID work) and proteins passed cut off in all the replicates were combined as our ERM proteome (150 proteins, 139 of them have secretory related GO annotations, 93% specificity). For the nucleus proteome, we abandoned the previous data and took four new replicates with 10 mM PA labeling (two with omitting blue LED irradiation as negative control, two with miniSOG-NES as negative control). H/L cut off was set by ROC analysis (list from TurboID work) and proteins passed cut off in all the replicates were combined as our ERM proteome (50

proteins, 46 of them have nucleus related GO annotations, 92% specificity). Although the coverage is relatively low of the ERM and nucleus proteome, the high spatial specificity of them indicates RinID as a powerful proximity labeling method to map subcellular proteomes (Figure 4A).

Figure 3C. Comparison of spatial specificity of RinID proteomic data with three other proximity labeling methods at mitochondria, Mitocarta 3.0 are defined as mitochondrial proteome.

Figure 4A. Comparison of spatial specificity of proteomic data derived from RinID and other proximity labeling methods at ERM and nucleus.

Minor points:

- The storyline of the paper - confusing back and forth between different probes used

Response: Thank you for pointing out this problem. We have repeated the experiments of *in vitro* and *in vivo* comparison of different probes. The new results are shown in the re-organized Figure S2. We have also double-checked the structures,

names and numbers of the probes throughout the text. The revised manuscript uses numbers instead of names of probes to improve readability and to remove ambiguity. For example, biotin-aniline is probe **2**, propargyl amine is probe **6**. The discussion about this part has also been re-organized in the Results section, paragraph 4, line 5-19: Meanwhile, we prepared a panel of biotin-conjugated amine probes (**1, 2, 4, 5**), including primary alkyl amine, aniline, and hydrazide, which differ in nucleophilicity, steric hindrance and basicity (Figure 2B). For comparison, we also included biotin-conjugated phenol (probe **3**), the commonly used substrate for APEX^{5, 6}. Since propargylamine has been used in CAP-seq to capture photo-oxidized guanosine, we added this probe (**6**) in our list of candidates as well. We tested the labeling efficiency of these probes both in the HEK293T cell lysate and live cells stably expressing cytoplasmic targeted miniSOG. For probe **6**, copper(I)-catalyzed alkyne-azide cycloaddition (CuAAC) click reaction with biotin-azide was performed to install the biotin moiety prior to Western blot analysis. Whereas a similar labeling efficiency was observed for probe **6** and probes **1-4** in the cell lysate (Figure S2A), only probe **6** (propargylamine) and probe **2** (biotin-aniline) yielded strong labeling signal in living cells (Figure S2B). We speculated that probes **1, 3, and 4** may have limited permeability through the cell membrane, while probe **5** lacks the ability to capture photo-oxidized protein intermediates.

Figure S2. Comparison of different probes for RinID activity in live cells and in cell lysates. **A)** Western blot and CBB of miniSOG labeled proteome samples with different nucleophilic probes in cell lysate; Probe concentration: 10 mM probe 1 and 6, 0.5 mM for other four probes. **B)** Western blot of miniSOG labeled proteome samples with different nucleophilic probes in miniSOG-NES HEK293T stable cell line. Probe concentration: 10 mM probe 1 and 6, 0.5 mM for other four probes. Bar graphs indicates the signal intensity of Streptavidin-HRP blot of each lane normalized by CBB. The concentration of SDS-PAGE is 4-20%.

- The mutated miniSOG version SOPP3, which is mentioned in the end, should have been used for the whole set of experiments.

Response: Thank you for this great advice. Indeed, SOPP3 or other better mutants of miniSOG could offer improved protein labeling efficiency which may lead to shorter labeling time and lower probe concentrations. We have decided to include miniSOG for the majority of the manuscript since this same protein has been used for photoactivatable protein labeling in CAP-seq. This means that the same miniSOG cell lines could be used for both transcriptomic profiling and proteomic profiling

experiments. For regular profiling experiments, miniSOG seems quite sufficient already, while SOPP3 is more needed for specialized experiments such as pulse-chase protein labeling, as we have demonstrated in HeLa cells. In the revised manuscript, we have highlighted the prior application of miniSOG in CAP-seq in the Introduction section, paragraph 7, line 1-4: **Notably, miniSOG has been used for mapping subcellular transcriptome (CAP-seq)¹⁹ and for probing protein-protein interactions²⁰, which demonstrates its high spatial specificity in labeling local biomolecules. Yet subcellular proteome-wide identification by miniSOG has not been reported.**

- In the text is a reference for Fig S1e-h but the Fig. S1 only has panels a-g

Response: Thank you for pointing out this mistake. Fig. S1 has been updated in the revised manuscript and the mistake has been corrected.

- Fig. 2d V5 band shows in miniSOG but does not show up in Fig. 2c. Additionally, V5-fusion should be mentioned in the figure description.

Response: Thank you for giving the advice. Fig. S2 has been updated in the revised manuscript. The fusion of V5-tag or HA-tag is now mentioned in the figure description of the figures containing V5 band or immunofluorescence channel (figure 2, figure S3, figure S4, figure S6, figure S7).

- Fig. S3b α -tubulin loading control varies drastically plus negative control is missing for some time points.

Response: Thank you for pointing out this confusion. We think the variation of the signal of α -tubulin is due to the problems in membrane transfer. Nevertheless, since the V5-tag signal is relatively uniform in the miniSOG containing samples, the intensity of SA-HRP blots of figure S3B (miniSOG contained groups only) has been normalized by V5 signal. The intensity plots are shown in figure S3 now.

Figure S3. Optimization of probe 6 concentration and illumination time. **A-B)** Western blot analysis of proteins labeled with mitochondrial matrix-targeted miniSOG (mito-V5-miniSOG) at different probe 6 concentrations (A) and illumination time (B). Bar graphs indicates the signal intensity of Streptavidin-HRP blot of each lane normalized by V5 (only miniSOG+ lanes are analyzed in (B)). The concentration of SDS-PAGE is 4-20%.

- Fig. S8b. Replicates differ quite drastically.

Response: Thank you for pointing out this point. We apologize for not explaining this more explicitly in the text. The polyacrylamide gel used for SDS-PAGE in Fig. S8b (figure S6B, top two images, in the revised manuscript) contain different percentages of acrylamide/bisacrylamide, which is why the protein bands in gel images appear different. The experiment of the left image used 12% SDS-PAGE, while the experiment of the right image used 4-20% concentration gradient SDS-PAGE. Nevertheless, these differences do not change the conclusions drawn from these experiments. In fact, our proteomic data have shown good reproducibility between replicates of labeling (Figure 4A-B, Table S3). In the revised manuscript, we have explicitly stated the differences in the polyacrylamide concentration used in these experiments in the explanatory text of each figure containing SDS-PAGE related results, to avoid confusing the readers.

REVIEWER COMMENTS

Reviewer #1 (Remarks to the Author):

The revised manuscript by Zheng et al. contains many substantial updates, experiments, clarifications, and text edits that largely satisfy the previous comments from this reviewer. This is overall a very nice work that strongly showcases a light-based proximity labeling method for profiling different cytosolic regions of the cell and will be useful to the scientific community and the proximity labeling field in general.

With that said, I have just three remaining concerns (not major) that should be addressed.

1) Figure 5 panel D. I like this figure layout but ultimately found it difficult to read. For example, Magenta is not easy to see in the plot. The authors should consider increasing the size of the dots, replacing magenta with another color (i.e. blue) and labeling some of the notable proteins called out in the text.

2) Also, regarding the interpretation of figure 5 panel D, and apologies if I missed this, but can the authors rule out or control for effects of SOPP3 + bluelight on this observed protein retention pattern. In other words, is it possible that the oxidation induced by this RinID treatment is contributing to this protein retention pattern or is it completely independent (or somewhere in between)? Knowing whether one is inducing this observed biology through this method or simply observing it independently is important for readers in considering how they might use this method. I realize that trying to figure this out might be beyond the scope of this initial work, however, some commentary in the text to address this potential perturbation effect or perhaps to qualify some of their observations should be helpful and sufficient.

3) Figure 4 panel C. Given the huge difference in RinID vs TurboID protein coverage, it might be helpful to also highlight that RinID was targeted to histone H2B and TurboID was more generally targeted to the nucleus somewhere in the figure or figure legend so there is some context to this big difference (in addition to only noting it in the text).

Reviewer #2 (Remarks to the Author):

While this revised manuscript is much improved, there are still a few arguable points regarding the pulse chase experiment. More critically, the miniSOG-based proximity labeling has recently been published by another group (Nat Commun 13, 4906 (2022)), which may lessen the novelty and originality of the authors' study due to the conceptual similarity. Thus, the authors may consider publishing to other journals and improving their work in the following points:

1. In the pulse-chase experiment, a discussion of the lifetime of each protein is missing. It is unclear whether the identified proteins are actually secreted or simply have a short lifetime. I am concerned that the estimated clearance speed may be an artifact, since proteins oxidized by the singlet oxygen-mediated labeling may be destabilized and have a shorter lifetime. To clarify this point, the rate of protein secretion and lifetime needs to be verified by other methods.

2. Why do some proteins show increased biotinylation levels after 8 hours? Does this mean that the labeling is still progressing even after light exposure because of the slow reaction kinetics between the amine probe and the oxidized protein?

Reviewer #3 (Remarks to the Author):

Fu Zheng, Chenxin Yu, Xinyue Zhou and Peng Zou did a great job in getting back to the reviewers. Thank you. The newly added RinID experiment showing the pulse-chase labeling of the ER proteome provides evidence that the technology can now be utilized to monitor subcellular proteotype dynamics - which is novel. Analysis of the data revealed a higher clearance rate for secreted proteins compared to ER resident proteins. These biological insights obtained from HeLa cells are not directly groundbreaking, but are sufficient as proof-of-concept and provide perspective for the design of future experiments in other cell systems. Together, RinID enables light-activated proximity-dependent protein labeling for profiling subcellular proteotypes which is great.

We thank all three reviewers for their thoughtful comments to help us further improve our manuscript. In this revision, we have provided additional experimental data and discussions regarding pulse-chase RinID labeling to improve data quality. We have also provided additional discussions comparing our work with others' that independently used miniSOG as photocatalytic proximity labeling tools. Please see our point-by-point responses below. These changes are marked as red in the revised text.

Reviewer #1:

The revised manuscript by Zheng et al. contains many substantial updates, experiments, clarifications, and text edits that largely satisfy the previous comments from this reviewer. This is overall a very nice work that strongly showcases a light-based proximity labeling method for profiling different cytosolic regions of the cell and will be useful to the scientific community and the proximity labeling field in general.

We thank the reviewer for these positive remarks.

With that said, I have just three remaining concerns (not major) that should be addressed.

1) Figure 5 panel D. I like this figure layout but ultimately found it difficult to read. For example, Magenta is not easy to see in the plot. The authors should consider increasing the size of the dots, replacing magenta with another color (i.e. blue) and labeling some of the notable proteins called out in the text.

Response: Thank you for this advice. We have updated Figure 5D accordingly to improve data readability. The updated figure is shown below.

Figure 5. D) Retention index of ER lumen proteins identified in pulse-chase RinID. Red: secreted proteins; Green: ER-resident proteins; Orange: ER resident with secreted subcellular location annotation.

2) Also, regarding the interpretation of figure 5 panel D, and apologies if I missed this, but can the authors rule out or control for effects of SOPP3 + blue light on this observed protein retention pattern. In other words, is it possible that the oxidation induced by this RinID treatment is contributing to this protein retention pattern or is it completely independent (or somewhere in between)? Knowing whether one is inducing this observed biology through this method or simply observing it independently is important for readers in considering how they might use this method. I realize that trying to figure this out might be beyond the scope of this initial work, however, some commentary in the text to address this potential perturbation effect or perhaps to qualify some of their observations should be helpful and sufficient.

Response: We thank the reviewer for this important advice. While it is experimentally difficult to completely rule out the possibility that the observed protein retention pattern may be affected by the oxidative RinID labeling, we note that the power and the time window of blue light irradiation used in pulse-chase RinID (30 mW·cm⁻² LED, 5 min) are substantially lower and narrower than those used in chromophore-assisted light inactivation (CALI: 540 mW·cm⁻² laser, 5 min; or 70 mW·cm⁻² laser, 25 min). Indeed, we did not observe significant changes in cell viability after 8 hours post-RinID labeling (Figure S8B). The fact that most proteins identified by pulse-chase RinID exhibited low degradation rate, which might be expected from extensive protein oxidative damage, also testifies that the labeling condition is mild on the protein targets (Figure 5C-D). The above evidence suggests that protein labeling by SOPP3 may not substantially affect the retention pattern of the identified proteins during the chase period. However, users of RinID should be cautious about the potential perturbation effect, particularly when using cell lines that are sensitive to oxidative stress. In the revised discussion, we have cautioned the readers of this issue.

The following discussion has been added to the revised manuscript:

Page 15, discussion section, paragraph 2.

In the pulse-chase RinID labeling, a critical question is whether the RinID labeling itself may affect the observed protein retention patterns. Although it is experimentally difficult to completely rule out such possibility, we note that the power and the time window of blue light irradiation used in pulse-chase RinID (30 mW·cm⁻² LED, 5 min) are substantially smaller than those used in CALI³⁶ (540 mW·cm⁻² laser, 5 min; or 70 mW·cm⁻² laser, 25 min). Indeed, cell viability was not significantly changed at 8 hours post-RinID labeling (Figure S8B), and most of the proteins identified by pulse-chase RinID did not exhibit high degradation rate (Figure 5C-D). The above evidence suggests that protein labeling by SOPP3 may not substantially affect the retention pattern of the identified proteins during the chase period. However, users of RinID

should be cautious about the potential perturbation effect, particularly when using cells that are sensitive to oxidative stress.

3) Figure 4 panel C. Given the huge difference in RinID vs TurboID protein coverage, it might be helpful to also highlight that RinID was targeted to histone H2B and TurboID was more generally targeted to the nucleus somewhere in the figure or figure legend so there is some context to this big difference (in addition to only noting it in the text).

Response: Thank you for this helpful advice. We have added this information to the legend of Figure 4C, please see below.

Figure 4. Subcellular proteomic profiling with RinID in the ER membrane (ERM) and nucleus. A) Comparison of spatial specificity of proteomic data derived from RinID and other proximity labeling methods at ERM and nucleus. B-C) Comparisons of proteomic coverage by RinID and other proximity labeling methods at the ERM (B) and nucleus (C). MiniSOG is fused with the histone protein H2B, which is targeted to chromatin. TurboID is fused with the nuclear localization sequence (NLS) and is targeted to the nucleoplasm.

Reviewer #2:

While this revised manuscript is much improved, there are still a few arguable points regarding the pulse chase experiment. More critically, the miniSOG-based proximity labeling has recently been published by another group (Nat Commun 13, 4906 (2022)), which may lessen the novelty and originality of the authors' study due to the conceptual similarity. Thus, the authors may consider publishing to other journals and improving their work in the following points:

1. In the pulse-chase experiment, a discussion of the lifetime of each protein is missing. It is unclear whether the identified proteins are actually secreted or simply have a short lifetime. I am concerned that the estimated clearance speed may be an artifact, since proteins oxidized by the singlet oxygen-mediated labeling may be destabilized and have a shorter lifetime. To clarify this point, the rate of protein secretion and lifetime needs to be verified by other methods.

Response: We thank the reviewer for these thoughtful comments. We agree with the reviewer that the observed clearance of proteins from the ER-lumen (as measured by retention index in the pulse-chase RinID experiment) could be caused by either protein secretion or degradation. Regarding the concern of RinID perturbing protein stability, we note that the time window and the power of blue light illumination used in pulse-chase RinID (30 mW·cm⁻² LED, 5 min) are much smaller than those used in CALI (chromophore-assisted light inactivation) experiments (540 mW·cm⁻² laser, 5 min; or 70 mW·cm⁻² laser, 25 min). Indeed, most of the proteins identified by pulse-chase RinID

did not exhibit high clearance rate at 8 hours post-labeling, and the cell viability was not significantly changed (Figure S8B). Together, the above evidence suggests that protein labeling by SOPP3 did not greatly affect the retention pattern of the identified proteins.

We have added the above discussion to the revised manuscript:

Page 15, discussion section, paragraph 2.

In the pulse-chase RinID labeling, a critical question is whether the RinID labeling itself may affect the observed protein retention patterns. Although it is experimentally difficult to completely rule out such possibility, we note that the power and the time window of blue light irradiation used in pulse-chase RinID ($30 \text{ mW} \cdot \text{cm}^{-2}$ LED, 5 min) are substantially smaller than those used in CALI³⁶ ($540 \text{ mW} \cdot \text{cm}^{-2}$ laser, 5 min; or $70 \text{ mW} \cdot \text{cm}^{-2}$ laser, 25 min). Indeed, cell viability was not significantly changed at 8 hours post-RinID labeling (Figure S8B), and most of the proteins identified by pulse-chase RinID did not exhibit high degradation rate (Figure 5C-D). The above evidence suggests that protein labeling by SOPP3 may not substantially affect the retention pattern of the identified proteins during the chase period. However, users of RinID should be cautious about the potential perturbation effect, particularly when using cells that are sensitive to oxidative stress.

As suggested by the reviewer, we have also sought to verify the measured rates of protein clearance in the ER lumen by an independent method. Metabolic incorporation of stable isotope-encoded amino acids (i.e. pulse-chase SILAC labeling) offers a non-perturbative approach to quantify protein turnover dynamics. To apply this method to analyze the sub-population of proteins in the ER lumen, one needs to combine pulse-chase SILAC labeling with subcellular-specific protein labeling. In another manuscript prepared by our group, titled *Spatially Resolved Mapping of Proteome Turnover Dynamics with Subcellular Precision* (NCOMMS-22-42858, under revision), we have developed one such technique (prox-SILAC) by incorporating enzyme-mediated proximity tagging into pulse-SILAC labeling scheme (see below).

In prox-SILAC, newly synthesized proteins within a specified time window (from 4 to 12 hours) are marked with heavy isotope-encoded lysine and arginine. Thereafter, proteins in the ER lumen are tagged with HRP via spatially restricted enzymatic labeling, enriched with affinity purification, and subsequently quantified via mass spec proteomic analysis. The turnover dynamics for a given protein can thus be quantified by calculating the ratio of heavy/light peak intensities from the mass spec data.

Thus, prox-SILAC offers another approach of quantifying protein retention with subcellular precision, albeit measuring from a slightly different angle from pulse-chase RinID. Both methods employ the idea of pulse-chase labeling. In prox-SILAC, peroxidase-mediated protein labeling occurs at the end of the chase period, offering a snapshot of mixed heavy and light protein populations in the ER lumen. In pulse-chase RinID, however, the protein labeling occurs prior to the chase, allowing the experimenter to follow the fate of marked protein sub-population through the chase period. We therefore do not expect these methods to give exactly identical results, but the overall trend should be similar. For example, proteins with very low ER retention rates as measured in pulse-chase RinID are expected to be characterized with high turnover rates as measured in prox-SILAC. For cell membrane proteins, high turnover in prox-SILAC and high retention in pulse-chase RinID are expected, as these proteins leave the ER lumen during the chase period but are still retained in the cell (i.e. not secreted).

We compared the protein turnover rate at 8 h in prox-SILAC (TR_{8h}) and the 8h/0h retention ratio (RR_{8h}) in pulse-chase RinID of the proteins identified in both methods (see the excel file 'pulse-chase RinID & prox-SILAC' attached to this response and the above figure). For most proteins, their TR_{8h} values are low and RR_{8h} values high, which are interpreted as strong ER retention. For example, the ER-resident protein P4HA1 (uniprot ID P13674) has $TR_{8h} = 0.259$ and $RR_{8h} = 1.098$. As one of the most rapidly

cleared proteins, LMAN2 (uniprot ID Q12907) exhibits high turnover dynamics in both experiments, with low $RR_{8h} = 0.629$ and high $TR_{8h} = 0.367$. Our analysis supports the high reliability of the protein retention pattern observed in pulse-chase RinID.

During the comparing, we noticed two cell membrane proteins with high both TR_{8h} and high RR_{8h} . ITGB1 (uniprot ID P05556, $TR_{8h} = 0.835$, $RR_{8h} = 0.954$) and NCSTN (uniprot ID O92542, $TR_{8h} = 0.776$, $RR_{8h} = 0.918$). The high retention of these two proteins in pulse-chase RinID indicates that they are stable in the cells after 8-hour chase, while their high turnover rates in prox-SILAC indicates that they are rapidly cleared from the ER lumen during the 8-hour chase. Such interpretation is consistent with the membrane trafficking of these two proteins. Thus, the comparison of pulse-chase RinID and prox-SILAC could provide a rich source of subcellular protein dynamics and transportation. We have included the prox-SILAC manuscript in this submission for the reviewers.

2. Why do some proteins show increased biotinylation levels after 8 hours? Does this mean that the labeling is still progressing even after light exposure because of the slow reaction kinetics between the amine probe and the oxidized protein?

Response: We thank the reviewer for pointing out this critical issue. Indeed, if the reaction between probe 6 and the oxidized histidine residue continues on even after turning off blue light irradiation, the temporal resolution of pulse-chase RinID could be severely undermined. To rule out such possibility, we designed the following experiments comparing the protein labeling intensities in SOPP3-KDEL HeLa stable cell line: 1) irradiating cells with blue LED in the presence of probe 6 for 5 min (normal labeling); 2) irradiating cells with blue LED in the absence of probe 6 for 5 min, followed by incubating cells with probe 6 in the dark for 5 min (staggered labeling); 3-4) negative controls where cells are kept in the dark for 5 min, in either the presence (3: omit BL) or the absence (4: omit BL and probe) of probe 6. Streptavidin blot analysis of protein biotinylation confirms that the staggered labeling sample (2) exhibits minimal labeling signal that is on par with the two negative controls (3-4), while the normal labeling sample (1) has substantially higher biotinylation signal. This demonstrates that probe 6 would not react with the oxidized proteins after the blue LED is switched off. This may be caused by the fast dynamic of the reaction between water and the oxidized residues or the dimerization of the oxidized residues. Thus, we attribute the increased biotinylation levels after 8 hours to the bias during sample preparation and MS identification.

The result of this experiment is shown in the newly added figure S8G:

Figure S8. G) Western blot to confirm whether probe 6 could react with the oxidized proteins after blue light irradiation. Cells were treated with four different conditions: (1) irradiated with blue LED in the presence of probe 6 in HBSS for 5 min (normal labeling); (2) irradiated with blue LED in the absence of probe 6 in HBSS for 5 min, followed by incubating with probe 6 in HBSS in the dark for 5 min (staggered labeling); (3) incubated with probe 6 in the dark in HBSS for 5 min (omit BL); and (4) incubated in HBSS in the absence of probe 6 in the dark for 5 min (omit BL and probe).

In the revised manuscript, we have added the following discussion:
Page 14-15, results section, paragraph 18.

It is worth noting that some proteins showed increased signal at 8 hours. To test whether protein labeling could continue to occur after the blue light irradiation is switched off, we designed the following experiments comparing the protein labeling intensities in SOPP3-KDEL HeLa stable cell line: 1) irradiating cells with blue LED in the presence of probe 6 for 5 min (normal labeling); 2) irradiating cells with blue LED in the absence of probe 6 for 5 min, followed by incubating cells with probe 6 in the dark for 5 min (staggered labeling); 3-4) negative controls where cells are kept in the dark for 5 min, in either the presence (3: omit BL) or the absence (4: omit BL and probe) of probe 6. Streptavidin blot analysis of protein biotinylation confirms that the staggered labeling sample (2) exhibits minimal labeling signal that is on par with the two negative controls (3-4), while the normal labeling sample (1) has substantially higher biotinylation signal (Figure S8G). This demonstrates that probe 6 would not react with the oxidized proteins after the blue LED is switched off. Thus, we attribute the increased retention index to the measurement errors introduced during MS quantitation.

Reviewer #3 (Remarks to the Author):

Fu Zheng, Chenxin Yu, Xinyue Zhou and Peng Zou did a great job in getting back to the reviewers. Thank you. The newly added RinID experiment showing the pulse-chase labeling of the ER proteome provides evidence that the technology can now be utilized to monitor subcellular proteotype dynamics - which is novel. Analysis of the data revealed a higher clearance rate for secreted proteins compared to ER resident

proteins. These biological insights obtained from HeLa cells are not directly groundbreaking, but are sufficient as proof-of-concept and provide perspective for the design of future experiments in other cell systems. Together, RinID enables light-activated proximity-dependent protein labeling for profiling subcellular proteotypes which is great.

Response: We appreciate the positive response of reviewer 3 to our revised manuscript.